



# COCAP: A carbon dioxide analyser for small unmanned aircraft systems

Martin Kunz[1], Jost V. Lavric[1], Christoph Gerbig[1], Pieter Tans[2], Don Neff[2], Christine Hummelgård[3], Hans Martin[3], Henrik Rödjegård[3], Burkhard Wrenger[4], and Martin Heimann[1,5]

[1]Max Planck Institute for Biogeochemistry, Jena, Germany
[2]NOAA Earth System Research Laboratory, Global Monitoring Division, Boulder, Colorado, USA
[3]SenseAir AB, Delsbo, Sweden
[4]Ostwestfalen-Lippe University of Applied Sciences, Hoexter, Germany
[5]Division of Atmospheric Sciences, Department of Physics, University of Helsinki, Finland

*Correspondence to:* Martin Kunz (mkunz@bgc-jena.mpg.de)

**Abstract.** Unmanned aircraft systems (UAS) could provide a cost-effective way to close gaps in the observation of the carbon cycle, provided that small yet accurate analysers are available. We have developed a COmpact Carbon dioxide analyser for Airborne Platforms (COCAP). The accuracy of COCAP's carbon dioxide ($CO_2$) measurements is ensured by calibration in an environmental chamber, regular calibration in the field and by chemical drying of sampled air. In addition, the package contains a lightweight thermal stabilisation system that reduces the influence of ambient temperature changes on the $CO_2$ sensor by two orders of magnitude. During validation of COCAP's $CO_2$ measurements in simulated and real flights we found a measurement error of $1.2\,\mu\text{mol}\cdot\text{mol}^{-1}$ or better with no indication of bias. COCAP is a self-contained package that has proven well suited for the operation on board small UAS. Besides carbon dioxide dry air mole fraction it also measures air temperature, humidity and pressure. We describe the measurement system and our calibration strategy in detail to support others in tapping the potential of UAS for atmospheric trace gas measurements.

## 1 Introduction

Atmospheric measurements of carbon dioxide ($CO_2$) are essential for our understanding of the carbon cycle and how it changes in a warming climate. Such measurements are made on a regular basis by global networks of surface stations, by specially instrumented aircraft and by research ships (Masarie and Tans, 1995). When local influences are filtered out, the data from these measurements allow the identification of global trends and the characterisation of major greenhouse gas sources and sinks, generally on the scale of continents (Fan et al., 1998; Ciais et al., 2010). This top-down approach to the quantification of the carbon cycle is well established.

In contrast, for observations on smaller scales conventional strategies often suffer from severe limitations. Specifically, the transition region between micro- and mesoscale in the sense of Orlanski (1975) poses a challenge. It comprises horizontal extents of 200 m to 20 km and periods from minutes to hours. Manned research aircraft do not fully cover this region due to minimum flight altitude requirements and their in most cases high airspeed. Furthermore, because their operation is costly,





they are typically deployed for short periods of time only. On the other side, stationary observations on instrumented masts or towers can deliver continuous data streams for long periods of time. However, they are fixed to a single location and take measurements from few vertical levels up to a maximum altitude limited by the height of the structure.

Unmanned aircraft systems (UAS), also called remotely piloted aircraft systems (RPAS), unmanned aerial vehicles (UAVs) or "drones", have the potential to fill this observational gap. Their use for research purposes has increased substantially over the past years. UAS that are capable of fully autonomous flight are now available for few thousand Euros, which has become possible by the development of small and cheap electronics for satellite navigation and inertial measurements as well as a growing consumer market. Especially smaller UAS with a mass of few kilograms are becoming more and more attractive for research. They have low system costs and can be operated by one or two persons. Obtaining an operating permission is easier for lightweight platforms and custom modifications do not require certification.

A fundamental limitation of small UAS is their payload capacity, both in space and mass. One reason why small UAS have been used in the field of meteorology for many years (e.g. Egger et al. 2002; Spiess et al. 2007; Reuder et al. 2008) is the availability of compact and lightweight instrumentation for the measurement of air temperature, humidity and pressure. Airborne studies of the carbon cycle, however, require accurate gas sensors, which are hard to miniaturise. Atmospheric signals on the micro- and mesoscale in $CO_2$ for example are typically in the range 1–100 $\mu mol \cdot mol^{-1}$, while the background $CO_2$ dry air mole fraction[1] $x_{CO2}$ is about 400 $\mu mol \cdot mol^{-1}$. If the sensitivity of a $CO_2$ sensor drifts by only 1 % during flight, e.g. due to the changes in ambient temperature, the resulting change by 4 $\mu mol \cdot mol^{-1}$ will obscure small signals.

Solutions for the measurement of greenhouse gases on board unmanned platforms have been found, but are not yet widely used, likely for practical or financial reasons. Berman et al. (2012) deployed a custom built laser-based $H_2O$, $CO_2$ and $CH_4$ analyser on the NASA SIERRA UAS, but the dimensions of $30 \times 30 \times 28$ cm$^3$ and the mass of 20 kg prevent the use of this analyser on smaller systems. Khan et al. (2012) developed a smaller laser-based analysers for $CO_2$ or $CH_4$ dry air mole fraction with a mass of 2 kg and a size of $20 \times 5 \times 5$ cm$^3$. The estimated drift in $x_{CO2}$ during 5–10 minute flights with a small helicopter was 1 %, which limits the system's suitability for environmental studies. Watai et al. (2006) deployed a 3.5 kg measurement package containing a nondispersive infrared $CO_2$ sensor on a UAS. They reported a comparably low bias of 0.21 $\mu mol \cdot mol^{-1}$ during tests with temperature changes similar to the conditions during flight. However, their setup requires 1 minute of in-flight calibrations every 6 minutes and comprises two gas cylinders, a pump and a drying cartridge in addition to a $20 \times 14 \times 8$ cm$^3$ main module.

Recently, attempts have been made to equip UAS with commercial-off-the-shelf $CO_2$ sensors designed for indoor air quality measurements. Their advantages are low cost, compact size and small mass of the measurement system. Brady et al. (2016) flew a 500 g payload containing such a $CO_2$ sensor on a small multicopter, but due to its high uncertainty (30 ppm plus 3% of reading according to manufacturer's specifications) the resulting data is hard to interpret. Numerous other groups have improved the accuracy of compact sensors by custom calibrations (e.g. Yasuda et al., 2012; Piedrahita et al., 2014; Shusterman et al., 2016; Martin et al., 2017). In some of these studies, measurement uncertainties below 5 $\mu mol \cdot mol^{-1}$ have been achieved

---

[1]Throughout this manuscript, $x_{CO2}$ denotes the $CO_2$ dry air mole fraction at a point, i.e. the result of an in situ measurement.





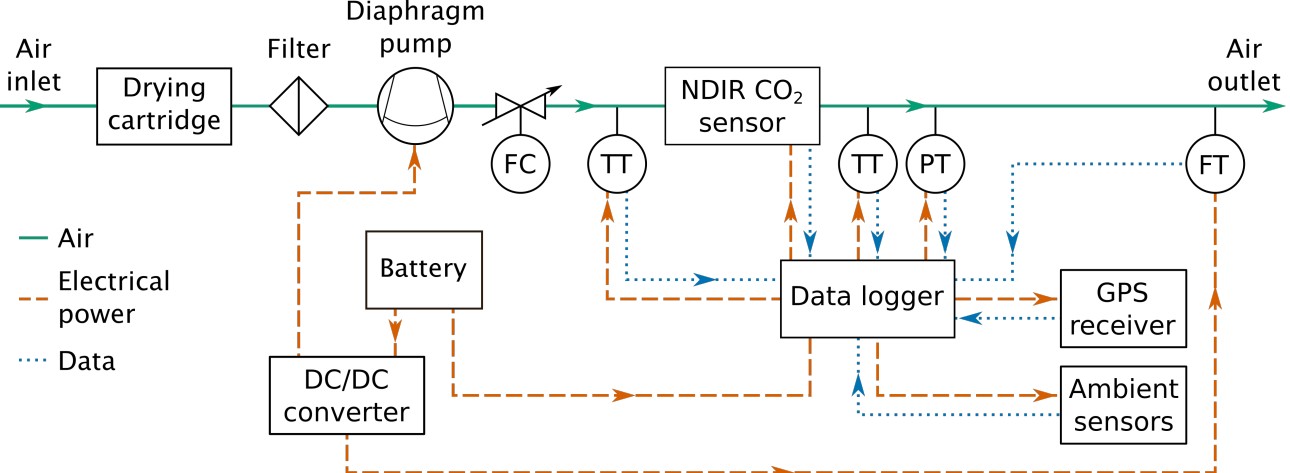

**Figure 1.** Flow of air, electrical power and data inside COCAP. FC - mass flow controller, TT - temperature transmitter (sensor), PT - pressure transmitter, FT - mass flow transmitter. The NDIR (nondispersive infrared) $CO_2$ sensor contains additional temperature sensors which are not included in this schematic view.

(Shusterman et al., 2016; Martin et al., 2017). However, none of these efforts aimed at the deployment of the sensors on UAS, which requires immunity to rapid changes in pressure and temperature as well as a high time resolution.

Aiming for a measurement package that is compact *and* accurate, we have developed COCAP, the COmpact Carbon dioxide analyser for Airborne Platforms. In Sect. 2 we provide a detailed description of our measurement system, consisting of (1)
5 COCAP, (2) a device that enables calibrations in the field and (3) an unmanned aircraft that carries COCAP. Section 3 focuses on our strategy to ensure accurate measurements through calibration of COCAP's sensors. In Sect. 4 we present the results from different tests that we carried out to assess COCAP's performance.

## 2   The measurement system

### 2.1   COCAP

10   COCAP measures $CO_2$ dry air mole fraction, temperature, relative humidity and pressure of ambient air. Furthermore, flow rate, pressure and temperature at different locations inside the analyser are recorded. We designed COCAP as an independent package containing not only sensors, but also control and data logging capabilities as well as a GPS receiver that provides position data and acts as time source. The mass of COCAP is 1 kg, excluding battery.

A schematic view of COCAP is provided in Fig. 1. The different components are described in the following subsections.





### 2.1.1 Carbon dioxide sensor

COCAP measures carbon dioxide using a sensor from SenseAir AB based on their HPP (High Performance Platform) family of gas sensors (Hummelgård et al., 2015). It is a nondispersive infrared sensor operating at a wavelength of 4.26 µm. The optical path has a length of 128 cm, which is obtained by 16 passes in an 8 cm long White cell (White, 1942). The total internal volume

of the cell is 48 cm$^3$. The mirrors are fabricated using plastic moulding which lowers the production cost. Heating elements and temperature sensors that enable temperature stabilisation of the optics are moulded into the mirrors. The sensor's mass is 80 g, including electronics.

### 2.1.2 Other components in the gas sampling line

Air that is drawn into COCAP's sample line is chemically dried as a first step. We use magnesium perchlorate (105873,

Merck KGaA, Germany) in cartridges built in-house from aluminium (Fig. S4). For improved drying performance we sieve the magnesium perchlorate and use only particles smaller than 2 mm. A single drying cartridge holds 1.5 g magnesium perchlorate, which is sufficient to dry nearly saturated air at a temperature of 24 °C and a flow rate of 300 ml·min$^{-1}$ to a water mole fraction of less than 200 µmol·mol$^{-1}$ for one hour.

Downstream of the drying cartridge a 0.2 µm filter (CM-0118, CO2Meter.com, USA) protects pump, flow regulator and

$CO_2$ sensor from particles. The air flow through the gas line is driven by a diaphragm pump (NMS 020 B, KNF Neuberger GmbH, Germany) and throttled to a flow rate of 300 ml·min$^{-1}$ by the mechanical flow controller (PCFCDH-1N1-V, Beswick Engineering Inc., USA). At this flow rate the pump reaches a differential pressure of 350 hPa, whereas the flow controller is preset at the factory to a higher pressure difference of 700 hPa. We lowered the setting of the flow controller, but tests indicate that in the current configuration the flow rate is directly proportional to ambient pressure, i.e. the flow controller behaves like

a needle valve. In future designs we recommend to better tune the flow controller for performance at low pressure differences or to reduce mass by replacing it with a needle valve.

The temperature of sample air is measured inside the inlet and the outlet of the $CO_2$ sensor with miniature thermistors (NCP15XH103F03RC, Murata Ltd., Japan) that are suspended from 0.2 mm diameter wire (Fig. S2 and S3). Pressure is measured with a compact piezoresistive pressure sensor (LPS331AP, STMicroelectronics, Switzerland). This model was chosen

for its small physical size, high resolution and digital interface. As it lacks a connection port, we glued a 3D-printed cap with a turned stainless steel port connector to the PCB so that it forms an air-tight enclosure around the pressure sensor (Fig. S5). The sample pressure is measured downstream of the $CO_2$ sensor close to its outlet. As the flow between the measurement cell and this point is virtually unrestricted, we assume equal pressure, i.e. we treat the reading of the pressure sensor as the pressure of the air inside the $CO_2$ sensor's measurement cell.

Finally, the mass flow rate of the sample air is measured with an analogue sensor (AWM3300V, Honeywell, USA). Downstream of the mass flow sensor the sample air is released from the gas line into COCAP's housing.





### 2.1.3 Ambient sensors

The measurement of features in temperature and humidity on the scale of tens of meters with UAS that can move several meters per second requires fast measurements with time constants on the order of 1 s. This calls for unrestricted or even forced ventilation of the sensing elements. To this end we designed a small ($60 \times 35$ mm$^2$) printed circuit board (PCB) that can be mounted in the most suitable location for any UAS (Fig. S6). Temperature is measured with a platinum resistance thermometer (Platinum 600 °C MiniSens Pt1000, IST AG, Switzerland), humidity is measured with a capacitive humidity sensor (P14 rapid in wired configuration, IST AG, Switzerland). Both sensors protrude over the edge of the PCB, which minimizes thermal mass and improves ventilation. They are protected from mechanical damage and contamination by an aluminium tube (length 30 mm, inner diameter 12 mm). The tube is polished on the outside to prevent heating by the sun and anodized matte black on the inside which avoids reflection and focussing of sunlight onto the sensors. On fixed-wing UAS we mount the PCB forward-facing, on rotary-wing UAS upward-facing in the downwash of the rotors.

### 2.1.4 Data logger

The data from all sensors are recorded to a memory card by a data logger. For this purpose we modified an electronics board designed for the operation with SenseAir's HPP gas sensors (Hök Instruments AB, Sweden) by adding connectors that provide an interface for the GPS receiver and different digital sensors. The board runs a firmware written for COCAP at the MPI for Biogeochemistry (Jeschag, 2014). Sensor data is recorded at 1 Hz. If a sensor samples at a higher rate, the measurements are averaged over one second. Data are continuously output via a serial interface and can be transmitted to a computer by means of an adaptor cable or a pair of radio modules (XBee 868, Digi International Inc., USA), allowing for real-time data visualisation and analysis. In addition, the data logger controls the built-in heaters of the $CO_2$ sensor to a user-adjustable temperature.

### 2.1.5 Temperature stabilisation

The temperature inside and outside COCAP influences the measurement of carbon dioxide in different ways. First, the density of the sample air and therefore the number of absorbing molecules in the measurement cell is inversely proportional to absolute temperature. Secondly, electronic components and the optical bandpass filter in the $CO_2$ sensor exhibit drift with temperature. Thirdly, the intensity of the absorption lines of any gas depends on temperature, which makes the optical depth of the sample air temperature dependent. Fourthly, changes in temperature influence the emission strength of the IR source. Fifthly, thermal expansion may cause mechanical deformations of the optical assembly. We correct the $x_{CO2}$ readings for drift with temperature (see Sect. 3.1), but experience with earlier setups shows that for best possible precision an active stabilisation of temperature is needed.

The HPP $CO_2$ sensor has built-in heaters and temperature sensors that are thermally coupled to the optical surfaces, the IR source and the optical detector. In an earlier version of COCAP we covered the $CO_2$ sensor with isolating material and preheated the air stream with a heated tube that was connected to the sensor inlet. In this setup, the temperature at three points could be precisely controlled to 50 °C, but the distribution of temperature was inhomogeneous and varying with ambient



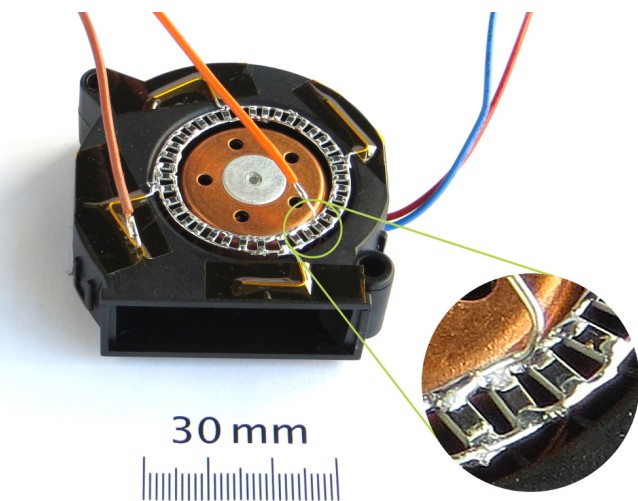

**Figure 2.** Fan and heating element used to stabilise temperature inside COCAP. The heating element is made of surface-mount technology resistors soldered to two concentric rings of wire. It is located just above the air inlet of the radial fan.

conditions. Although a total of 9 temperature sensors were present at different locations in COCAP, the uneven temperature distribution made it impossible to fully determine the system's thermal state and a satisfactory temperature correction could not be established.

Consequently, we redesigned the stabilisation system to minimize temperature differences around the $CO_2$ sensor. This goal
requires a setup where the heat exchange between different parts of the sensor happens much faster than dissipation of heat from the sensor to the changing environment. In many instruments this is achieved by means of massive bodies of copper or aluminium, which are characterised by high thermal diffusivity. However, as the mass of large metal parts is unacceptable for our application, we use air to transport heat inside COCAP. The lower thermal diffusivity of air is compensated for by forced convection driven by a fan. A heater, a temperature sensor and a custom PCB are connected and programmed as a
control loop that stabilises the air temperature at 50 °C. The warm air stream circulates throughout COCAP's housing (Fig. S1 in supplement) so that not only the $CO_2$ sensor is decoupled from changes in ambient temperature, but all electronics boards benefit from the stabilisation as well.

While being flown on a UAS the temperature around COCAP can change by several degrees within seconds. Compensation of these changes requires a fast control loop, which calls for a heating element and a temperature sensor with low thermal mass.
We built the heating element (Fig. 2) from 38 surface-mount technology resistors (nominal resistance 360 $\Omega$, length$\times$width $2\times1.25$ mm$^2$, rated power 0.5 W) which are connected in parallel and provide a maximum heating power of 15 W at 12 V. We installed the heating element just above the air inlet of the fan (RLF 35-8/12 N, ebm-papst Mulfingen GmbH & Co. KG, Germany). Placing the heating element at the outlet of the fan would reduce the response time, but likely result in a less homogeneous temperature distribution across the air stream.



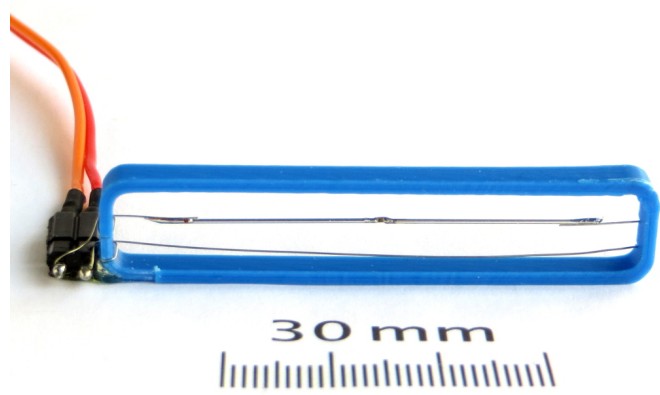

**Figure 3.** Sensor used for the measurement of the temperature of the air circulating inside COCAP. The sensing element (located in the centre of the blue 3D-printed frame) is a miniature thermistor ($1\,\text{mm}\times0.5\,\text{mm}\times0.5\,\text{mm}$). Two 23 mm long pieces of 0.5 mm wire are soldered to both sides of the thermistor. They conduct heat, facilitating the measurement of a temperature that is averaged across the air stream.

The temperature sensor ("air stream sensor", Fig. 3) in the control loop is a miniature thermistor (NCP15XH103F03RC, Murata Ltd., Japan). It is placed close to the detector of the $CO_2$ sensor and oriented perpendicular to the flow of circulating air to minimize flow resistance.

Fan, heating element and air stream sensor are connected to an in-house-built control board (Fig. S7 and S8) that runs a PID
(proportional–integral–derivative) controller implemented in software. The temperature-dependent resistance of the air stream sensor is measured in a Wheatstone bridge configuration with a resolution of 1 mK at 50 °C. A trimmer potentiometer allows calibration of the temperature measurement. The control board detects if the thermistor is shorted or disconnected and disables heating in such cases.

The power dissipation of the heating element is controlled by pulse-width modulation with 16 bits resolution. The control
board monitors the current drawn by the fan and powers the heating element only if the fan is operating normally. This protection prevents damage to the heating element which would overheat if the fan is disconnected or broken.

The performance of the temperature stabilisation under flight conditions is discussed in Sect. 4.1.2 and Sect. 4.2.2.

### 2.1.6 Battery

COCAP requires a source of electrical power with a voltage in the range 10.2 V–14.0 V. We use a single lithium polymer
battery with a nominal voltage of 11.1 V. The battery life time depends on the ambient temperature, but generally we can operate COCAP for more than one hour from a 2 200 mAh battery weighing 200 g. By choosing a different battery size the package can be optimised for lower mass or longer run time.



## 2.2 Field calibration device

In order to make the handling of gas standards in the field easier and safer we developed a field calibration device (Fig. S9). It consists of a gas cylinder dolly, accommodating two 10 l aluminium cylinders, and a valve box. The cylinders are securely tied to the dolly so that they can stay in place at all times, which simplifies transportation. Inside the valve box a three way valve

allows switching between the two gas standards or shutting off the gas stream. The flow rate is controlled to $400\,ml\cdot min^{-1}$ with a mechanical flow controller (PCFCDH-1N1-V, Beswick Engineering Inc., USA). The connection to COCAP has an open-split configuration so that there is always an $100\,ml\cdot min^{-1}$ overflow and the gas standard is delivered at ambient pressure. At a field site three steps are necessary to make the system ready for use: (1) removing the protective caps from the cylinders, (2) mounting (including leak-checking and flushing) of the pressure regulators (Model 14, Air Liquide USA LLC) and (3)

connecting them to the valve box. For details of the field calibration sequence and gas standards see Sect. 3.3

## 2.3 Unmanned aircraft system

COCAP has no dependence on any external system and can therefore be deployed on a variety of UAS. The only requirements are a payload capacity of typically 1.2 kg (depending on the choice of battery, see Sect. 2.1.6) and space for the $14\times14\times42\,cm^3$ package, either inside the fuselage or attached to the outside of the vehicle. Sufficient ventilation must be ensured to prevent

overheating. In Germany, UAS can be operated without a permit at low heights up to 100 m above ground level if their take-off mass is lower than 5 kg. COCAP's size and mass generally allow to meet this limit. We have operated COCAP on multicopters and carried out successful tests on a fixed-wing aircraft.

### 2.3.1 Multicopters

Multicopters are rotorcraft with more than two rotors. They are generally easier to handle than fixed-wing aircraft due to

their hovering ability and their built-in electronic control systems. Multicopters feature vertical take-off and landing as well as arbitrarily low vertical and horizontal flight speed. However, for aerodynamic reasons they typically have a lower endurance than fixed-wing aircraft of similar mass. Moreover, the strong mixing around and below the rotors gives rise to an uncertainty of the origin of a measured air sample. We alleviate this problem by sampling air through a carbon fibre tube with the inlet placed sideways or above the rotors.

So far we have flown COCAP on two different multicopters. One is an octocopter (S1000, DJI Ltd., China) with a total take-off mass of 9.2 kg. The other one is a quadcopter (custom-built, Sensomotion UG, Germany and Ostwestfalen-Lippe University of Applied Sciences, Germany, Fig. S10) weighing 4.8 kg with COCAP mounted. Both multicopters are electrically powered and provide at least 10 minutes of flight time.

### 2.3.2 Fixed-wing aircraft

Fixed-wing aircraft require a minimum air speed to fly and generally depend on an airstrip or additional equipment like bungees and nets for take-off and landing. However, they tend to have higher endurance and longer reach than multicopters. We carried



out successful flight tests with an electrically powered fixed-wing aircraft (X8, Skywalker Technology Ltd., China) using a dummy that has the same mass and size as COCAP. The complete system weighed approximately 3.6 kg.

## 2.4 Cost estimation

The cost estimate provided here includes materials and components in the state in which we procured them, but excludes any labour associated to their modification and assembly. We estimate the material costs for COCAP at EUR 4 500 and for the field calibration device (including two cylinders, but not the gas standards inside) at EUR 3 300. The recurring costs for gas standards and drying agent are few euros per flight and thus negligible.

Commercial off-the-shelf multicopters with a payload capacity of at least 1.2 kg are available for EUR 3 000, including essential equipment such as battery, charger and remote control.

## 3   Calibration

### 3.1   Calibration curve of the $CO_2$ sensor

A nondispersive infrared gas sensor measures the fraction of one component in a mixture of gases utilising the characteristic absorption bands that many substances exhibit in the infrared. The HPP sensor inside COCAP outputs a signal that is approximately proportional to the intensity of infrared radiation that has passed through the gas mixture. Absorption by one constituent of the gas reduces the intensity, but the relation between the mole fraction of this component and the intensity is non-linear and depends on temperature and pressure of the gas. Furthermore, sensor elements like the infrared source and detector can have a temperature dependence that influences the measurement. Generally, the carbon dioxide mole fraction $x_{CO2}$ of the gas mixture can be calculated as

$$x_{CO2} = f(s, T_G, p_G, \ldots) \tag{1}$$

where $s$ is the infrared signal, $T_G$ and $p_G$ are temperature and pressure of the gas mixture, respectively, and the ellipsis indicates that other quantities may be included in the calculation. The function $f$ will henceforth be called the "calibration curve" of the carbon dioxide sensor.

Although an ab initio calculation of the calibration curve is in principle possible, we did not follow this approach due to lack of information (e.g. the precise transmission characteristics of the optical bandpass filter). Instead, we made a series of measurements with known $CO_2$ dry air mole fraction under changing ambient conditions and used regression analysis to approximate the calibration curve. To this end we placed COCAP in an environmental chamber where ambient temperature and pressure could be varied over the range expected in field deployments (Fig. 4 panel b and c). Temperature was changed in a step pattern from 28 °C to 0 °C while pressure was smoothly adjusted from 1100 hPa to 700 hPa and back during each temperature step. These disparate patterns were chosen to ease the attribution of sensitivities to one of the independent variables. Sample air with gradually changing $CO_2$ dry air mole fraction was provided to COCAP from a spherical, stainless steel, 8 l buffer volume



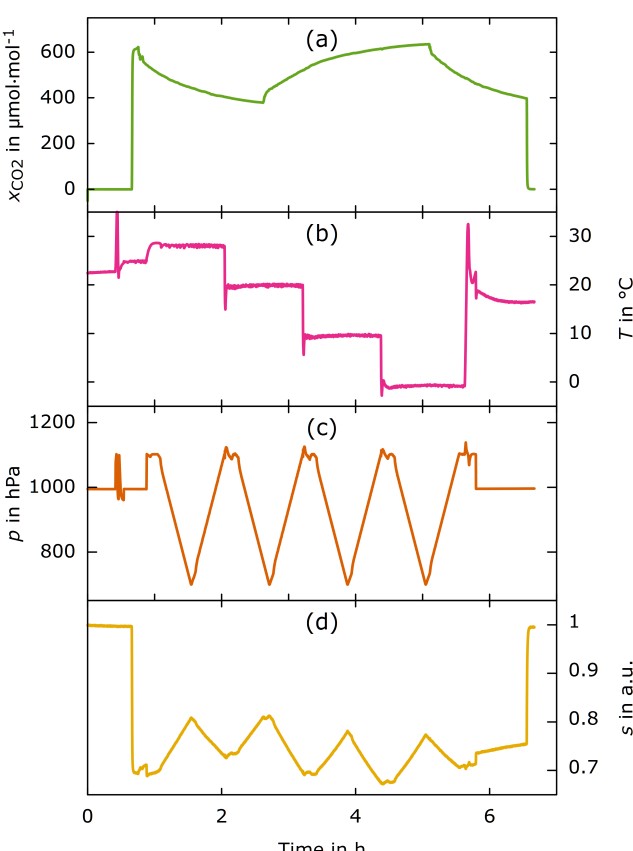

**Figure 4.** Conditions during one of the experiments carried out to define a calibration curve for the $CO_2$ sensor: (a) $CO_2$ dry air mole fraction $x_{CO2}$ measured with a reference analyser, (b) ambient temperature $T$, (c) ambient pressure $p$. (d) Normalised infrared signal $s$ from the optical detector of COCAP's $CO_2$ sensor. The variations in pressure influence the observed signal more strongly than the changes in $x_{CO2}$ by $200\,\mu\text{mol}\cdot\text{mol}^{-1}$, which illustrates the need for a precise pressure correction.





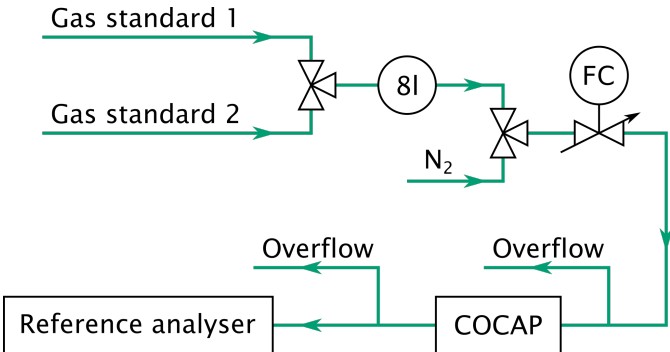

**Figure 5.** Setup used to determine a calibration curve for the $CO_2$ sensor. FC is a mass flow controller. The 8 l buffer volume was flushed with either of two gas standards differing in $CO_2$ content, providing a slowly changing $CO_2$ dry air mole fraction. The second three-way valve is used to deliver nitrogen with zero $CO_2$ at the beginning and the end of each experiment.

that was continuously flushed with air from one of two gas cylinders (Fig. 5). One cylinder contained a lower-than-ambient $CO_2$ dry air mole fraction (349.9 μmol·mol$^{-1}$), the other one was enriched with $CO_2$ (648.6 μmol·mol$^{-1}$). At a flow rate of 400 ml·min$^{-1}$ the buffer volume acted approximately as a first order low-pass filter with a time constant of 20 min (Fig. 4a). Air leaving COCAP was directed to a Picarro G2401 cavity ring-down spectroscopy (CRDS) analyser (Anthony O'Keefe and

David A. G. Deacon, 1988) that had been calibrated to the WMO $CO_2$ X2007 scale and served as reference. Open splits upstream and downstream of COCAP ensured that sample air was delivered to both analysers at chamber pressure despite different flow rates (COCAP 300 ml·min$^{-1}$, Picarro G2401 200 ml·min$^{-1}$).

The dominant features in the infrared signal (Fig. 4d) are the step changes in $x_{CO2}$ and the influence of pressure changes. Note that in a non-dispersive infrared sensor the signal is inversely related to the amount of absorbing molecules in the measurement

cell. Hence, both low $x_{CO2}$ and low pressure lead to a high signal. The gradual changes in $x_{CO2}$ by 200 μmol·mol$^{-1}$ between 0:40 h and 6:30 h have a smaller influence on the infrared signal than the changes in pressure do. This shows the importance of a precise correction for ambient influences.

In total we carried out three experiments similar to the one depicted in Fig. 4 on consecutive days. The variation in temperature and pressure were the same for all experiments, but the initial $CO_2$ dry air mole fraction in the buffer volume differed and

the switching between low-$CO_2$ and high-$CO_2$ cylinders took place at different times.

Regression analysis of the experimental data was carried out in GNU Octave using the *leasqr* function from the *optim* package. It is an implementation of the Levenberg–Marquardt algorithm for non-linear least-squares regression that allows the variation of some parameters of a model while others are fixed. This capability enabled a stepwise approach in which we fitted one part of a model at a time until finally all parameters could be set free. In contrast, straightforward fitting of a complete

model at once did not converge. We attribute this to the high number of parameters (10 or more) and to the lack of initial values sufficiently close to the optimum.

The models that we fitted to the experimental data are of the form



$$
\begin{aligned}
x_{CO2} \quad &= \quad f(s,T,p...) & (2) \\
&= \quad \frac{g_1(T_{Inlet},T_{Outlet})}{g_2(p)} \cdot g_3(g_4(s,...)) + c & (3)
\end{aligned}
$$

The function $g_1$ represents the inlet temperature, $g_2$ the pressure inside the measurement cell of the $CO_2$ sensor. The fraction $g_1/g_2$ originates from the ideal gas law, but in many of our models it has a more general form, including constant and quadratic

terms to account for other effects like temperature or pressure broadening of absorption lines. The function $g_3$ is a polynomial of second or third order that approximates the non-linear relation between amount of absorbing molecules in the measurement cell and light intensity at the detector. Finally, the purpose of $g_4$ is to correct the detector signal for disturbances, e.g. gain drift with temperature.

In our regression analysis we repeatedly performed three steps: (1) formulation of a model, (2) fitting of the model to the

experimental data by minimising the sum of squared residuals and (3) evaluation of the model performance. Given the large number of parameters in the models, the second step was susceptible to "overfitting", i.e. fitting to a point at which the model represents not only an underlying process, but also random variations in the experimental data. We countered overfitting in two ways. On the one hand, we rejected models in which an additional parameter reduced the sum of squared residuals only insignificantly. On the other hand, we validated the fitted models against independent data measured on a different day, which

allowed us to assess the stability of the parameters over time. Based on these considerations we chose the following calibration curve:

$$
\begin{aligned}
x_{CO2} \quad &= \quad \frac{a_1 T_{Outlet} + a_2}{p + a_3}(a_4 g_4^3 + a_5 g_4^2 + g_4 + a_6) + a_7 & (4) \\
g_4(s,p) \quad &= \quad \frac{s + a_8}{a_9 p + a_{10}} & (5)
\end{aligned}
$$

$a_1$ through $a_{10}$ are the parameters fitted in the regression. The capability of this calibration curve to compensate for ambient

influences is illustrated in Fig. 6.

### 3.2 Calibration of ambient sensors

We calibrated COCAP's temperature, humidity and pressure sensors prior to measurements in the field. The general calibration approach was the same for all three sensor types: In a first step, we placed them together with a reference instrument in an environmental chamber, varied the relevant ambient conditions and recorded the indications of both the sensor under test

and the reference. In a second step, we fitted a model that relates the indications of the sensor under test to the reference measurements. This model can later be used to correct sensor indications during field measurements.

Calibration of the temperature and humidity sensors was carried out in a PSL-2KPH chamber (ESPEC Corp., Japan). A chilled-mirror dew point hygrometer (Michell Dewmet TDH) with a measurement uncertainty of $0.1\,°C$ for temperature and $0.2\,°C$ for dew point served as the reference. Reference and sensor under test were placed close to each other and actively

ventilated during the measurements.



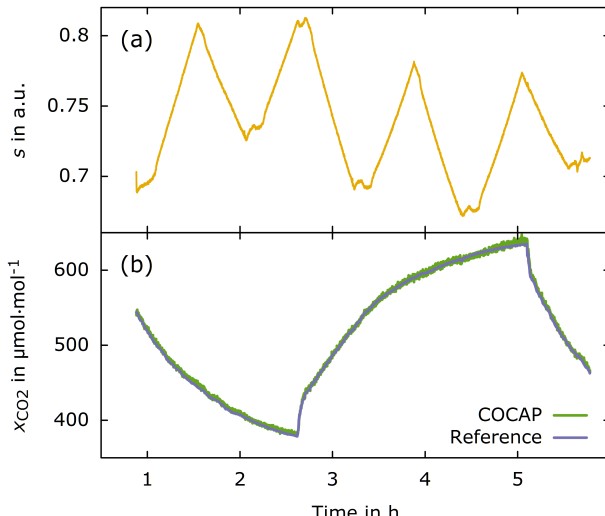

**Figure 6.** (a) Normalised infrared signal of COCAP's $CO_2$ sensor under changing ambient conditions. This is a subset of the data shown in Fig. 4d. (b) $x_{CO2}$ calculated using the calibration curve (Eq. 4 and 5) compared to the measurements of a reference analyser (Picarro G2401). The $x_{CO2}$ signal is recovered despite strong influences from changing pressure.

**Table 1.** Summary of the calibration of COCAP's ambient sensors: Conditions during calibration, correction model and root mean square error (RMSE) of the corrected indications with respect to the reference. $T$, $U$ and p are uncorrected temperature, relative humidity and pressure, respectively. $T_C$ is the corrected air temperature.

| Sensor | Temperature in °C | Relative humidity in % | Pressure in hPa | Model | RMSE |
|---|---|---|---|---|---|
| Temperature | 5–40 | 50 | 1000 | $a_1 T^2 + a_2 T + a_3$ | 0.04 °C |
| Humidity | 10–30 | 15–100 | 1000 | $a_1 U^2 + a_2 U + a_3 + a_4 U \cdot T_C + a_5 T_C$ | 1.4 % |
| Pressure | 0–40 | N/A | 400–1000 | $a_1 p^2 + a_2 p + a_3$ | 1.1 hPa |





The pressure sensors were calibrated in a CH3030 chamber (SIEMENS AG, Germany). The reference instrument (Druck DPI 740, General Electric Company, USA) has a measurement uncertainty of 0.26 hPa.

Details of the calibrations are listed in Table 1. The correction model for the humidity sensor depends not only on the raw humidity signal, but also on air temperature. It therefore relies on the corrected indication of the temperature sensor.

A potential source of error in the humidity calibrations are the different response times of the chilled-mirror dew point hygrometer and the slower reacting capacitive humidity sensor. We avoid the introduction of a bias by calibrating with slowly varying symmetric humidity patterns, i.e. by using the same rate of change during humidity increase and decrease.

### 3.3 Field calibration

The $CO_2$ dry air mole fraction reported by COCAP drifts over time (see Sect. 4.1.1), necessitating periodic calibration. We
decided against in-flight calibrations to reduce the system mass, to save space and to have the full flight time available for the measurement of ambient air. Instead we sample two gas standards before and after each flight using the field calibration device described in Sect. 2.2. One of the standards has a $CO_2$ dry air mole fraction close to clean ambient air (397.57 $\mu$mol·mol$^{-1}$), the other one is enriched with $CO_2$ (447.44 $\mu$mol·mol$^{-1}$). Both standards consist of natural air collected at the Max Planck Institute for Biogeochemistry in Jena, Germany, i.e. the standards are similar in isotopic composition to the air that we typically measure
in the field. This way we avoid isotope related errors that can occur with synthetic air standards (Tohjima et al., 2009). The standards are sampled for 5 minutes each. We discard the first 3 minutes to ensure that the measurement system is well flushed, leaving 2 minutes for averaging. This time span is a compromise between noise reduction (the minimum standard error of the mean would be achieved by averaging over 4 minutes, see Sect. 4.1.1) on the one hand and consumption of gas standards and time spent for the field calibration (and lost for ambient measurements) on the other.

During data analysis, the calibration curve (see Sect. 3.1) is applied to all measurements of COCAP, resulting in a time series of $CO_2$ dry air mole fraction. Averaging is carried out for each gas standard and each sampling period by calculating the arithmetic mean of the $CO_2$ dry air mole fraction. Next, "virtual" standard measurements are created by interpolating between the calculated averages linearly in time. Using these virtual standard measurements two correction parameters $a(t)$ (slope) and $b(t)$ (offset) are calculated for each point in time between the first and last standard measurement such that the difference
between corrected measurement and assigned value vanishes. The corrected $CO_2$ dry air mole fraction is thus calculated as

$$x_{\text{CO2}} = a(t) \cdot f(s, T, p...) + b(t) \tag{6}$$



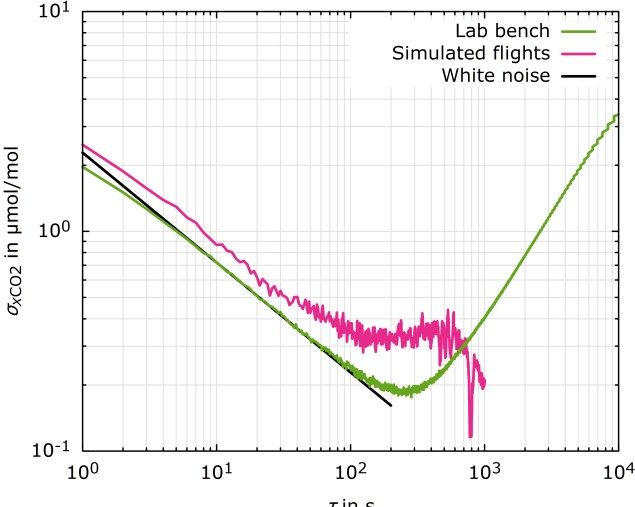

**Figure 7.** Allan deviation of unfiltered $CO_2$ dry air mole fraction $\sigma_{x\mathrm{CO2}}$ versus averaging period $\tau$. Air from a cylinder was measured under laboratory conditions and during the simulated flights described in Sect 4.1.2. The latter is depicted until $\tau = 1000\,\mathrm{s}$, i.e. one sixth of the length of the time series. The characteristic -1/2 slope of white noise is indicated by the black line.

## 4 Performance tests

### 4.1 In the lab

#### 4.1.1 Allan deviation of $CO_2$ dry air mole fraction

The influence of white noise on a measurement can be reduced by averaging over several samples. However, the precision
achievable by averaging is limited by drift of the instrument over time, i.e. over long averaging periods the imprecision caused by drift becomes larger than the imprecision caused by noise. Allan or two-sample variance $\sigma_y^2(\tau)$ is a measure commonly used to analyse these two sources of error. It is defined as (Allan, 1987)

$$\sigma_y^2(\tau) = \frac{1}{2}\left\langle (\Delta y)^2 \right\rangle \tag{7}$$

where $\Delta y$ is the difference between two consecutive averages over a period of $\tau$ and the angle brackets denote the expected
value. The square root of the Allan variance is called Allan deviation $\sigma_y$.

To characterise noise and drift of COCAP we connected the analyser in an open split configuration to a cylinder containing natural air with a $CO_2$ dry air mole fraction of $384.3\,\mu\mathrm{mol}\cdot\mathrm{mol}^{-1}$. This air sample was measured in a lab environment over a period of 24 hours. The Allan deviation of this dataset (Fig. 7) reaches a minimum of approximately $0.2\,\mu\mathrm{mol}\cdot\mathrm{mol}^{-1}$ at an averaging period of $230\,\mathrm{s}$. For averaging periods shorter than $100\,\mathrm{s}$ the curve has a slope of $-\frac{1}{2}$ in the log-log plot, which is
characteristic of white noise. At $\tau_C = 1800\,\mathrm{s}$, the typical time between field calibrations, the Allan deviation is approximately





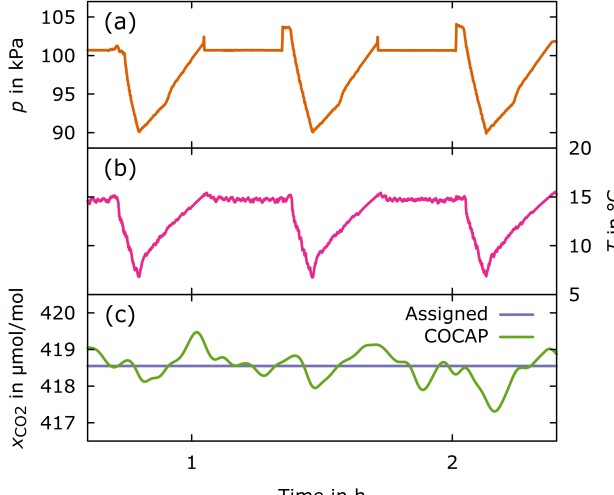

**Figure 8.** Stability test in an environmental chamber. The variations in ambient pressure $p$ and temperature $T$ approximately represent three flights between 0 and 1000 meters above sea level in the International Standard Atmosphere. Air with constant dry air mole fraction of $CO_2$ was supplied to COCAP from a cylinder. Calibrations were carried out before and after the flights (not shown) using two gas standards ($397.5\,\mu\text{mol}\cdot\text{mol}^{-1}$ and $447.4\,\mu\text{mol}\cdot\text{mol}^{-1}$). The dry air mole fraction $x_{CO2}$ measured with COCAP has been smoothed by convolution with a Gauss window of 200 s full width at half maximum in order to reduce the high-frequency noise. The sharp features in pressure were caused by turning the environmental chamber's pressure regulation off after and back on before each flight.

$0.7\,\mu\text{mol}\cdot\text{mol}^{-1}$ and dominated by drift. Our correction scheme (see Sect. 3.3) removes offset and gain errors at the point in time when the field calibrations were carried out and uses linear interpolation between the calibrations. Therefore, the largest uncorrected drift is expected to be lower than $0.4\,\mu\text{mol}\cdot\text{mol}^{-1}$, the Allan deviation for $\frac{1}{2}\tau_C = 900\,\text{s}$. Figure 7 also shows the Allan deviation of COCAP's $x_{CO2}$ measurements during the simulated flights described in Sect. 4.1.2. Forced ventilation and

5 substantial changes in ambient pressure and temperature during the simulated flights lead to an increased Allan deviation compared to the measurements in a lab environment. Estimation of the expected drift for $\frac{1}{2}\tau_C = 900\,\text{s}$ is not feasible due to artefacts that become visible at averaging periods beyond 400 s.

### 4.1.2 Simulated flights

An instrument that is flown on a small UAS can be exposed to rapid changes in temperature and pressure, especially if the

10 flight pattern covers a large range in altitude. To assess the measurement error caused by such changes, we simulated three consecutive flights between 0 and 1000 m above sea level (Fig. 8 panels a and b) in an environmental chamber (CH3030, SIE-MENS AG, Germany). Temperature and pressures were controlled to approximately resemble the International Civil Aviation Organization Standard Atmosphere (ICAO, 1993). Each simulated flight had a duration of 20 minutes with 5 minutes ascent and 15 minutes descent. After a 20 minutes break without changes in temperature and pressure this pattern was repeated. Only



**Table 2.** Mean $\overline{T}$, standard deviation $\sigma$ and range $T_{max} - T_{min}$ of temperatures under simulated flight conditions.

|            | $\overline{T}$ in °C | $\sigma$ in mK | $T_{max} - T_{min}$ in mK |
|------------|------|------|------|
| Ambient    | 12.83 | 2366 | 8786 |
| Air stream | 50.00 | 4    | 79   |
| Inlet      | 50.25 | 10   | 50   |
| Outlet     | 50.12 | 32   | 110  |
| Detector   | 54.86 | 10   | 50   |

dry air samples from cylinders were supplied to COCAP, hence no drying cartridge was necessary. Otherwise, the analyser was operated in standard configuration, including pump and flow control. An open split with overflow upstream of the analyser's inlet ensured that air was sampled at the pressure of the environmental chamber at all times. First we measured two gas standards with $CO_2$ dry air mole fractions of 397.5 µmol·mol$^{-1}$ and 447.4 µmol·mol$^{-1}$ for 5 minutes each at 15 °C and 100 kPa.

Afterwards, air with 418.6 µmol·mol$^{-1}$ $CO_2$ was supplied over a period of two hours while the environmental chamber was simulating three flights as detailed above. Finally, we sampled the two gas standards again for 5 minutes each at 15 °C and 100 kPa. The $x_{CO2}$ measurement by COCAP is affected by different sources of error: random noise, drift over time, calibration errors and drift with temperature and pressure. We reduced the noise by convoluting the time series with a Gauss window of 200 s full width at half maximum (Fig. 8c). A two-point calibration was derived from the standard measurements at the begin-

ning and end of the test and applied to the full time series using linear interpolation in time. This cancelled out linear drift over time, but due to the influence of noise on the measurement of the gas standards and non-linearity in the instrument response a calibration error remains. Drift with temperature and pressure is corrected for with the calibration curve described in Sect. 3.1, but this correction does not completely eliminate the effect of these parameters on the measurement result. The $x_{CO2}$ time series in Fig. 8c exhibits a local minimum whenever pressure and temperature are minimal, with a maximum deviation from

the assigned value of the cylinder (418.6 µmol·mol$^{-1}$) of -1.2 µmol·mol$^{-1}$ at 2:10 h. The bias of the mean over the whole test, representing the combined effect of the calibration error and drift with temperature and pressure, was $-0.03$ µmol·mol$^{-1}$. The standard deviation of the 1 Hz time series before convolution with a Gauss window was 2.7 µmol·mol$^{-1}$, the standard deviation after convolution was 0.41 µmol·mol$^{-1}$.

   Figure 9 shows time series of temperatures measured at different points inside COCAP during the simulated flights. Statistics

of these time series are given in Table 2. The differences in the observed patterns result from the air circulation inside COCAP's housing: First, heated air from the fan streams along the inlet tube of the $CO_2$ sensor. Next, it passes the air stream sensor (see Sect 2.1.5). Finally, it reaches the outlet tube and the detector of the $CO_2$ sensor.

   The air stream sensor is part of the control loop and therefore its temperature stays close to the set point of 50 °C. The inlet and outlet temperature sensors measure the temperature of the sample gas, which indirectly reflects the temperature of

the air circulation because heat is exchanged through the walls of the inlet and outlet tubes. The temperatures of the outlet





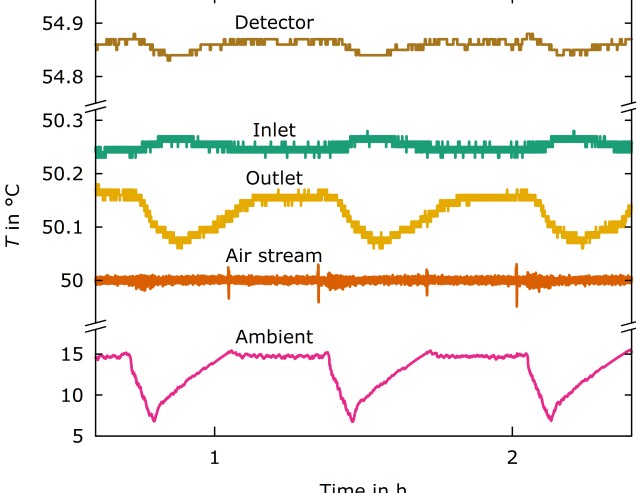

**Figure 9.** Temperatures inside and outside COCAP under simulated flight conditions (depicted in Fig. 8 panels a and b). The temperature sensors in the detector, inlet and outlet of the $CO_2$ sensor are uncalibrated, hence they have an unknown offset of up to 1.1 °C. The detector is about 5 K warmer than the other parts (note the broken temperature axis) due to heat transfer from the component block which is heated to 55 °C. Ambient temperature is plotted at a different scale as it varies two orders of magnitude more than the stabilised internal temperatures. The spikes in air stream temperature are related to fast pressure changes in the environmental chamber (see Fig. 8a).

exhibits minima whenever the ambient temperature is reduced. This is caused by increased heat transfer from the circulating air to outside COCAP's housing. The inlet shows the inverse behaviour, i.e. increasing temperature under reduced ambient temperature, because it is located between fan and air stream sensor. In an colder environment the air has to be heated more to keep the temperature at the air stream sensor constant, so the temperature at the inlet rises.

5     The temperature of the detector varied qualitatively similar to that of the outlet because detector and outlet are located close to each other. However, the temperature of the detector is approximately 5 K higher and the amplitude of the temperature variations is roughly three times lower than at the outlet. This is due to the detector's thermal coupling to the component block, which is temperature controlled to 55 °C. Overall, the temperature stabilisation reduces the variability in the internal temperatures by two orders of magnitude compared to the changes in ambient temperature (see Table 2).

10   **4.2   Field deployment**

**4.2.1   Measurements with an instrumented van**

Testing COCAP under realistic conditions requires the measurement of $x_{CO2}$ in ambient air, i.e. in air with changing humidity and $CO_2$ content. A reference instrument must sample the same air to enable the evaluation of COCAP's performance. Laser-based instruments are widely used for high-accuracy measurements of greenhouse gases (e.g. Laurent, 2016) and therefore





suitable to be used as a reference, but at a mass of more than 10 kg they are too large to fly on any UAS available for this work. As an intermediate step between stationary indoor testing and UAS flights we integrated COCAP into the "Mobile Lab" (Pétron et al., 2012), an instrumented van equipped with a CRDS analyser for carbon dioxide, carbon monoxide, methane and water (G2401-m, Picarro Inc., USA). Air was sampled at 3.5 m above street level. Field calibrations (see Sect. 3.3) were carried

out by injecting air from cylinders with slightly higher-than-ambient pressure into the sampling line at regular intervals.

As COCAP and the CRDS analyser were connected to the same sampling line, delay and mixing caused by tubing and inlet filter affected them equally. However, the flushing time for the analyser's measurement cells is different, which makes direct comparison of their readings inappropriate. In the following we explain how we handled this issue mathematically.

The flushing process can be described as a convolution of the $CO_2$ dry air mole fraction at the inlet of the sampling line,

$x_{\mathrm{Inlet}}(t)$, with an analyser-specific instrument function $f(t)$:

$$x(t) \quad = \quad (x_{\mathrm{Inlet}} * f)(t) \tag{8}$$

$$= \quad \int_{-\infty}^{\infty} x_{\mathrm{Inlet}}(t - t') \cdot f(t')dt' \tag{9}$$

The response $x(t)$ of the analyser is the reported $CO_2$ dry air mole fraction. Due to causality, $x(t)$ cannot depend on future $CO_2$ dry air mole fractions at the inlet. Hence, the lower limit of the integration can be set to zero:

$$x(t) = \int_{0}^{\infty} x_{\mathrm{Inlet}}(t - t') \cdot f(t')dt' \tag{10}$$

The instrument function $f(t)$ is not known *a priori*, but can be estimated from the response to a step change in $x_{\mathrm{Inlet}}(t)$. Such step changes occurred at the end of calibration measurements when the supply of gas standard into the sampling line was shut off. From the data we found that for both analysers the response $x_{SC}(t)$ to a step change can be modelled by an exponential decay of the form

$$x_{SC}(t) = (x_0 - x_\infty) \cdot e^{-t/\tau} + x_\infty \tag{11}$$

where $x_0$ and $x_\infty$ are the $CO_2$ dry air mole fractions before and after the step change, respectively, and $\tau$ is the characteristic time constant of the flushing process. We determined time constants of 13 s for COCAP and 25 s for the CRDS analyser. To find the function $f(t)$ we differentiate Eq. 10:

$$\frac{d}{dt}x(t) \quad = \quad \frac{d}{dt}\int_{0}^{\infty} x_{\mathrm{Inlet}}(t - t') \cdot f(t')dt' \tag{12}$$

$$= \quad \int_{0}^{\infty} f(t') \cdot \frac{d}{dt}x_{\mathrm{Inlet}}(t - t')dt' \tag{13}$$



In case of a step change at the inlet, the differentiation yields the Dirac delta function $\delta$, scaled by the height of the step $(x_\infty - x_0)$:

$$\frac{d}{dt}x_{SC}(t) = \int_0^\infty f(t') \cdot (x_\infty - x_0) \cdot \delta(t - t')dt' \tag{14}$$

$$= (x_\infty - x_0) \cdot f(t) \tag{15}$$

Rearranging and applying Eq. 11:

$$f(t) = \frac{\frac{d}{dt}x_{SC}(t)}{x_\infty - x_0} \tag{16}$$

$$= \frac{(x_\infty - x_0) \cdot e^{-t/\tau}}{(x_\infty - x_0) \cdot \tau} \tag{17}$$

$$= \frac{e^{-t/\tau}}{\tau} \tag{18}$$

This means that the instrument function of either analyser can be described with an exponential decay which has the same time constant as the analyser's step response. For practical reasons, we treat $f(t)$ as equal to zero outside $0 \le t \le 5\tau$. Because no carbon dioxide is created or removed inside the analysers, the time integral over $f(t)$ must be equal to one, necessitating normalisation of the truncated response function:

$$f'(t) = \begin{cases} 0 & \text{if } t < 0 \\ \frac{f(t)}{1 - e^{-5}} & \text{if } 0 \le t \le 5\tau \\ 0 & \text{if } t > 5\tau \end{cases} \tag{19}$$

Through the measurement process both analysers effectively convolute the $x_{CO2}$ signal present at the inlet of the sampling line with their respective instrument function. To make the results comparable, we convolute the measurements of each analyser with the instrument function of *the other* analyser. If both measurements were perfect, this would yield identical results because convolution is commutative:

$$x_{COCAP} * f_{CRDS} = (x_{Inlet} * f_{COCAP}) * f_{CRDS} \tag{20}$$

$$= (x_{Inlet} * f_{CRDS}) * f_{COCAP} \tag{21}$$

$$= x_{CRDS} * f_{COCAP} \tag{22}$$

Here $x_{COCAP}$ and $x_{CRDS}$ are the $CO_2$ dry air mole fractions measured by COCAP and the CRDS analysers, respectively. Convoluting each analyser's measurement with the instrument function of the other analyser can therefore be viewed as an equalisation of the flushing times of both analysers.




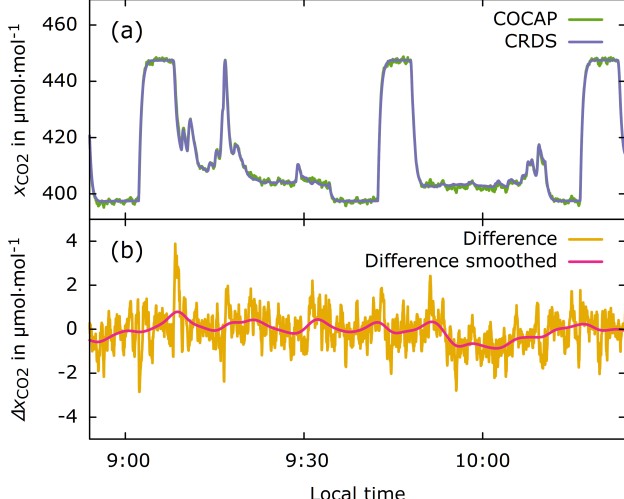

**Figure 10.** (a) CO₂ dry air mole fraction $x_{CO2}$ of ambient air measured by COCAP and a CRDS analyser during a car drive. The three step-like patterns originate from the field calibration during which two gas standards are sampled. The peak at 9:16 occurred while waiting at a traffic light on a busy street. (b) Difference and smoothed difference of $x_{CO2}$ measured by COCAP minus $x_{CO2}$ measured by the CRDS analyser. The smoothing is implemented by convolution with a Gauss window of 200 s full width at half maximum. The measurements of both analysers have been corrected using the field calibrations and the flushing times have been equalised as explained in the text.

We carried out the experiment with the Mobile Lab in Boulder, Colorado, USA, on 15 October 2015. We drove up and down a road that covers 700 m in elevation (1650–2360 m above sea level), exposing COCAP to the same pressure changes that would occur during a flight at these altitudes. The temperature inside the van increased from 14 °C to 20 °C during the drive. Despite the substantial changes in ambient conditions, the measurements from both analysers agree to within 2 µmol·mol⁻¹
during most of the experiments (Fig. 10). Differences are mainly caused by limitations of the simple model for the instrument functions, which become apparent during fast changes in $x_{CO2}$, and by sensor noise. They are reduced to less than 1 µmol·mol⁻¹ when high-frequency variations are filtered out. The negative bias of about -0.8 µmol·mol⁻¹ observed between 9:55 and 10:15 might be the effect of sunlight exposure. At the time of the experiment, the temperature stabilisation described in Sect. 2.1.5 had not yet been implemented, which is why COCAP was sensitive to changes in ambient conditions.

### 4.2.2 Lannemezan flights

The so far highest flight of COCAP on a UAS was carried out in Lannemezan, France, on 20 May 2016 at 15:30 UTC (17:30 local time). COCAP was mounted under a custom-built multicopter (Sensomotion UG, Germany and Ostwestfalen-Lippe University of Applied Sciences, Germany, Fig. S10). Starting from an elevation of 600 m above sea level a maximum height of 430 m above ground level was reached. The flight took place under clear sky in a light breeze (2 m/s wind speed). It served as a real-world test of COCAP's temperature stabilisation (Fig. 11). While ambient pressure $p$ changed by 5 kPa and ambient





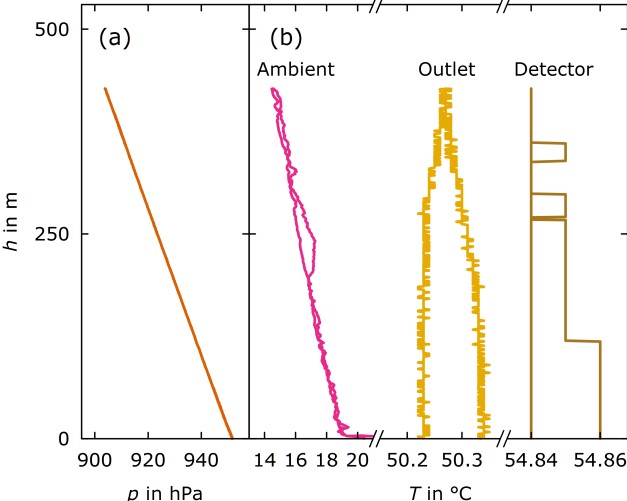

**Figure 11.** Stability of COCAP's internal temperatures during a flight to a maximum height of $h = 430\,\mathrm{m}$ above ground level. $p$ is ambient pressure, $T$ is temperature. Note the broken temperature axis and the different scaling used for ambient, outlet and detector temperature.

temperature $T_{\mathrm{Ambient}}$ by $4.5\,°\mathrm{C}$ during the flight, the temperature of the air stream at the $CO_2$ sensor's outlet $T_{\mathrm{Outlet}}$ varied by only $0.13\,°\mathrm{C}$ and the temperature of the optical detector $T_{\mathrm{Detector}}$ by only $0.02\,°\mathrm{C}$.

### 4.2.3   Comparison to the atmospheric observatory Lindenberg

In order to verify COCAP's in-flight measurements of ambient $CO_2$ dry air mole fraction, we made a comparison to the at-
mospheric observatory Lindenberg. The Lindenberg observatory is part of the Integrated Carbon Observation System (ICOS, www.icos-ri.eu) and meets the World Meteorological Organization (WMO) Global Atmosphere Watch (GAW) recommenda-tions for high accuracy atmospheric trace gas measurements (WMO, 2016). The Lindenberg observatory (ICOS short name: LIN) is located in the eastern part of Germany (52°10' N, 14°07' E) in a flat, rural area. At LIN a 99 m mast air is equipped with inlets at 2.5 m, 10 m, 40 m and 98 m above ground level. Air is drawn from all inlets continuously, but only one sampling line is
analysed at a time. The gas analyser is switched to a different sampling line every five minutes ("quasi-continuous sampling"). It measures carbon dioxide and methane dry air mole fraction.

We mounted COCAP on the same multicopter that was used in Lannemezan (see Sect. 4.2.2) and carried out a total of 21 flights close to the mast (distance less than 150 m) on 18 and 27 October 2016, using the same setup on both days. During each flight, the multicopter ascended vertically to 100 m above ground level at a climb rate of $0.5\,\mathrm{m \cdot s^{-1}}$, followed by a descent at a
rate of $2\,\mathrm{m \cdot s^{-1}}$. The same pattern was then repeated at approximately 70 m distance from the first ascent. Each flight lasted 9 minutes and the two ascents were separated in time by 4 minutes. The only exception was flight 7, which we had to abort after the first ascent due to a technical problem with the multicopter.





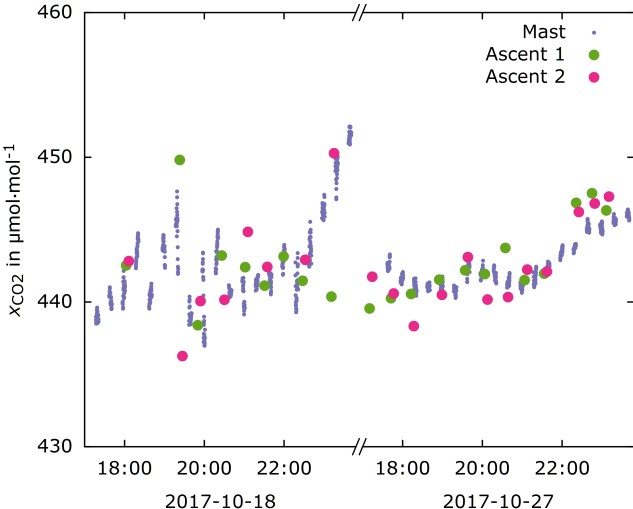

**Figure 12.** Large dots represent calibrated measurements taken by COCAP during the ascents at a height of $(40\pm10)\,\mathrm{m}$ above ground, small dots are measurements from the LIN mast's 40 m inlet. Times are in local time (UTC+2). The sun set at 18:02 local time on 18 October and at 17:43 local time on 27 October. The gaps in the station data are due to the measurement of other inlets and working tanks at those times.

In flight, the multicopter mixes the air around it. To avoid artefacts in our measurements caused by this mixing, we sampled air from 70 cm above the rotors. Furthermore, for the analysis detailed below we used only the data collected during the ascents, i.e. when the UAS was flying into and sampling undisturbed air. In each flight, we carried out the second ascent upwind of the first ascent, which ensured that the mixing caused by the first ascent did not degrade the measurements during the second

ascent.

On the evening of 18 October, the sky was cloudy with occasional drizzle, the wind low $(1.5\,\mathrm{m\cdot s^{-1}})$, and the lowest 100 meters of the atmosphere were weakly stable. On the early evening of 27 October, conditions were similar, but without precipitation. After 21:00 local time on 27 October, the sky cleared up, followed by the formation of radiation fog.

At the mast's 2.5 and 10 m inlets, the variability of the $CO_2$ dry air mole fraction was larger than above due to respiration

fluxes from soil and vegetation that were intermittently mixed upwards by turbulence. This high variability makes these levels less suitable for comparison to COCAP, as our flights were not synchronised with the sampling at the mast. We were not allowed to fly higher than 100 m, hence the 98 m inlet was not well covered by the flight pattern. We therefore focus our comparison on the 40 m inlet of the mast.

After applying the calibration curve to the measurements of COCAP's $CO_2$ sensor, we corrected for its temporal response

by deconvolution. For each ascent, we then calculated the arithmetic mean of the measurements taken between a height of 30 and 50 m. Figure 12 shows these means together with the LIN measurements at 40 m plotted against time. Overall, there is good agreement between COCAP and LIN. The variability in COCAP's data is higher, likely caused by a low-pass effect of the mast's sampling system. Further differences are due to the measurements being taken 100 m–150 m apart and at different





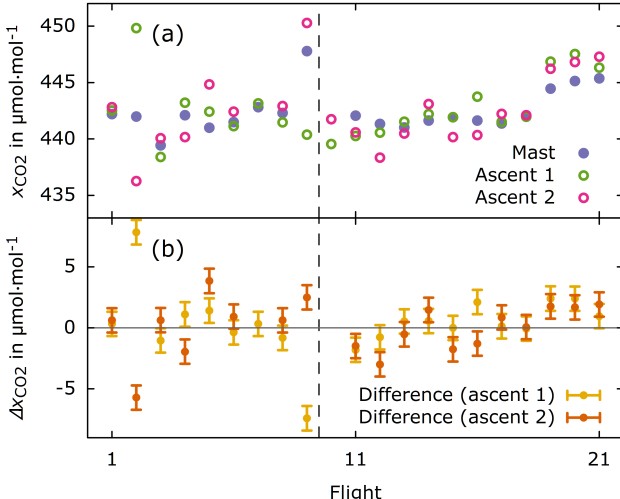

**Figure 13.** (a) COCAP data are the same as in 12. Measurements from the mast's 40 m inlet have been averaged over a period that starts 20 min before and ends 20 min after the respective flight. No measurements from the mast are available for flight 10 because calibration cylinders have been measured at this time. (b) $x_{CO2}$ measured by COCAP minus $x_{CO2}$ measured by LIN. Error bars indicate the noise level of COCAP's $CO_2$ sensor after deconvolution. Flights 1 through 9 were carried out on 18 October, flights 10 through 21 on 27 October (indicated by dashed vertical line).

times. Finally, instrument noise leads to discrepancies. All these factors should have a zero mean effect, whereas a consistent bias between COCAP and the Lindenberg observatory would indicate a problem with the calibration. A change of bias over time would suggest instrument drift.

To better assess the presence of bias, we averaged the measurements from the mast's 40 m inlet over a period starting 20 min before and ending 20 min after the respective flight (Fig. 13a). We then calculated the difference between the measurements by COCAP and LIN (Fig. 13b). Additionally, we estimated the noise level of COCAP's $CO_2$ sensor. The correction for COCAP's finite time response by deconvolution has the side effect of amplifying high frequency electronic noise. Therefore, we did not use the Allan deviation of the data described in Sect 4.1.1, but calculated the Allan deviation of the deconvoluted time series. During each ascent, the multicopter climbed from 30 to 50 m height in approximately 40 s. In analysing a period during which a standard gas was measured, we found an Allan deviation of 1 μmol·mol$^{-1}$ for $\tau = 40$ s. This number is represented as error bars in Fig. 13b.

Table 3 contains the mean difference between the measurements by COCAP and LIN. A bias of zero lies within one standard error of the mean difference. This is also true for both nights considered individually and for both ascents of all flights considered individually. Table 4 lists the results of statistical tests of three hypotheses:(1) no bias between COCAP and LIN, (2) no change in the mean difference between COCAP and LIN from 18 October to 27 October and (3) no change in the mean





**Table 3.** Mean of the difference between COCAP's $CO_2$ dry air mole fraction measurements $x_{COCAP}$ and the corresponding measurements by LIN $x_{LIN}$ ($\pm$ 1 standard error). Subsets of COCAP's measurements are also analysed.

| COCAP measurements | $\overline{x_{COCAP} - x_{LIN}}$ in µmol·mol$^{-1}$ |
|---|---|
| All | $0.23 \pm 0.45$ |
| 18 October | $0.16 \pm 0.85$ |
| 27 October | $0.28 \pm 0.49$ |
| All first ascents | $0.39 \pm 0.66$ |
| All second ascents | $0.06 \pm 0.52$ |

**Table 4.** Statistical tests for bias. Here $x$ denotes measurements by COCAP and the index defines a subset: 'A' for all measurements, '18' and '27' for 18 and 27 October, respectively, and 'A1' and 'A2' for first and second ascent, respectively. $D(x)$ represents the difference between $x$ and the corresponding measurements by LIN. An overline denotes the arithmetic mean.

| Null hypothesis | Statistical test | Test result $p$ |
|---|---|---|
| $\overline{D(x_C)} = 0$ | Welch's $t$-test | 0.75 |
| $\overline{D(x_{18})} = \overline{D(x_{27})}$ | Welch's $t$-test | 0.72 |
| $\overline{D(x_{A1})} = \overline{D(x_{A2})}$ | Student's $t$-test | 0.75 |

difference between COCAP and LIN from the first to the second ascends. None of the hypotheses was rejected ($p > 0.7$ in all cases).

We conclude that the measurements gave no indication of (1) calibration problems, (2) uncorrected drift of COCAP between 18 and 27 October or (3) drift of COCAP during flight.

## 5  Summary and conclusions

With COCAP we have designed and built a self-contained analyser for the measurement of $CO_2$ dry air mole fraction, temperature, humidity and pressure of ambient air on board UAS. COCAP is typically operated under ambient conditions that change quickly and over wide ranges. These challenging conditions can compromise the accuracy of $CO_2$ sensors. We ensure COCAP's accuracy by (1) temperature stabilisation, (2) drying of sample air, (3) a calibration curve that includes correction terms for temperature and pressure and (4) by regular field calibrations. When high-frequency noise is filtered out, COCAP's measurements of $CO_2$ dry air mole fraction were found to deviate from a reference by not more than $1.2\,$µmol·mol$^{-1}$ during



simulated flights and by not more than $1\,\mu mol\cdot mol^{-1}$ during deployment in an instrumented van. In a comparison to the ICOS observatory Lindenberg no indication of bias or uncorrected drift was observed.

Since the design of COCAP, newer versions of SenseAir's HPP sensor family have become available. They exhibit lower drift and lower noise at a slightly smaller form factor (Arzoumanian et al., 2016). The integration of the newer sensors into COCAP would be straightforward and is expected to further improve the accuracy of the $x_{CO2}$ measurements.

With a volume of $14\times14\times42\,cm^3$and a mass of $1\,kg$ COCAP fits onto small UAS with a take-off mass below $5\,kg$. It is therefore a cost-effective tool to study carbon dioxide in the lowest 100–1000 m of Earth's atmosphere. On a multicopter or fixed-wing aircraft COCAP enables measurements at a finer scale than manned aircraft and without restrictions of minimum flight altitude. On a tethered balloon, COCAP can take measurements for longer time spans without being bound to fixed altitudes like an instrumented mast or tower.

The techniques presented in this article are applicable to other measurement systems as well. Many sensors benefit from a stable temperature and we have shown how an effective temperature stabilisation can be achieved within the mass, size and power restrictions of a small UAS. Likewise, the presented method for obtaining a calibration curve can be applied to other gas sensors. Regular calibrations are important to ensure the accuracy of trace gas measurements and we have given an example how to implement them in a practicable way.

Flying a $CO_2$ analyser on small UAS opens up new possibilities in studying the carbon cycle. As a first application we have constrained nocturnal carbon dioxide fluxes using repeated $x_{CO2}$ profiles in a budget method (Kunz et al., in preparation). Other potential uses include measuring the emission strength of point sources and investigation of emissions in urban areas.

Due to their unique capabilities and low cost, we foresee that the use of unmanned aircraft in the Earth sciences will significantly increase in the near future. We have shown how accurate measurements of the $CO_2$ dry air mole fraction can be taken on board small UAS and we anticipate these platforms to play an important role in closing gaps in the observation of the carbon cycle.

*Data availability.* The analyses presented here are based on many different experiments and use a combination of two or three different data sources in most cases. Compiling the data into a uniform, self-describing collection suitable for upload to a repository would be a great effort. Given the fact that our experiments were aimed at characterising COCAP, reuse of the data by other groups seems unlikely. Hence, we did not upload our measurement data to a repository. However, data from individual experiments is available from the corresponding author upon request.

Data from the ICOS station Lindenberg can be requested from ICOS-D (http://www.icos-infrastruktur.de/en/mitarbeiter/atmosphaerenprogramm/).

*Competing interests.* Christine Hummelgård, Hans Martin and Henrik Rödjegård work for SenseAir AB, the manufacturer of the HPP sensor family. The other authors declare that they have no conflict of interest.

*Acknowledgements.* We thank Maksym Bryzgalov (SenseAir AB, Sweden) for helping with the configuration and integration of the $CO_2$ sensor. Hök Instruments AB (Sweden) kindly provided software for COCAP's data logger. We thank Wieland Jeschag and Till Fastnacht for adapting this software to our needs. We acknowledge Jürgen Kaulfuß, who designed and built the field calibration device. We thank Frank Beyrich, Matthias Lindauer, Udo Rummel and Marcus Schumacher (Deutscher Wetterdienst, Germany) for access to the Lindenberg station, technical support and data sharing. We gratefully acknowledge the authors of various open-source software packages that were used in the project and for the preparation of the manuscript, in particular GNU Octave (including the *optim* package), KiCad, gnuplot, GIMP, Inkscape, LyX and LaTeX. We thank the Max Planck Society for generous financial support. Parts of this work have been funded by COST (European cooperation in science and technology) and the German academic exchange service DAAD.





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
