# Peer review of "COCAP: A carbon dioxide analyser for small unmanned aircraft systems"

_Atmospheric Measurement Techniques, 2017_

## Referee Comment (RC1) · Anonymous Referee #1 · 22 Nov 2017

General Comments: The text describes a portable CO2 observing device designed primarily for use on small unmanned aerial vehicles. The authors take a prototype NDIR device from a commercial gas sensor company and include additional hardware and sensors to enhance performance and test it on unmanned aircraft platforms as well as on a van in Colorado. Methodologies for calibration both in the lab as well as in the field, both including reference gases, are described in detail. It was found that air pressure plays a significant role in the reported value and must be accounted for in a calibration. A thorough set of evaluations were performed in a controlled laboratory setting, concurrently with a laser-based analyzer in a mobile experiment, and flying alongside the inlets of a high-accuracy tower. Through these evaluations, it was found that the developed sensing platform can make observations of sufficient quality for

many scientific applications.

Specific Comments: Pg. 1, line 20: key word is "fully" cover, there are "missed approaches" for research aircraft where they can attempt a landing and get a vertical profile very close to the surface.

The sections on the various types of UAVs seems irrelevant to the paper. Consider saving yourself the cost of ~1 page and remove or shorten this section to just list them.

During the airborne test, interference from turbulence is accounted for. Is there any potential for interference from vibrations from the aircraft (the mirrors in the CO2 sensor perhaps?)? Could you examine this easily and if so, how would one account for this in your instrument design?

Sect. 4.2.1: Just to clarify, the drying of the inlet air is still performed on the COCAP, right? The first sentence with "changing humidity" may be a bit confusing, consider changing/rewording.

Technical Comments: Pg. 2, line 19: CO2 is defined earlier, but here CH4 is written without stating "methane (CH4) earlier (and H2O as well, since it is water vapor specifically). It's generally good practice to define future abbreviations.

---

## Referee Comment (RC2) · Anonymous Referee #2 · 4 Dec 2017

General Comments:

An interesting paper that nicely describes a creative way for the use of low cost measurements with a satisfying number of technical details. After addressing the comments below the manuscript will be suitable for publication.

Specific Comments:

It would be interesting to see in the conclusion more future directions that focus on different scientific questions that could be answered with this measurement method. Based on the paper the main purpose of this measurement method is "close gaps in the observation of the carbon cycle", however maybe try expanding this statement. You refer to this in the introduction and conclusion but maybe highlight more what is and

isn't possible to do with this measurement method. Explain more which gaps you are trying to close and why.

Similarly in the conclusion line 25 you state 'Other potential uses include measuring the emission strength of point sources and investigation of emissions in urban areas', yes this is possible with your measurement method but it is also possible with numerous other instruments. It does not emphasise a unique benefit of this instrument. Also you wouldn't actually measure the emission strength directly, but make measurements that can be used to investigate the strength of point sources.

It would be good to add a table with the approximate height, area, time that can be covered when they use the UAS (with details about the specific setup and when they take into account everything e.g. battery life etc) relative to other platforms described in the paper.

Technical corrections:

Change the x and y labels on the plots - Time in h change to Time (h) or better to Time / h

Section 2.1.5 is there a reference for all the 5 statements of how temperature affects CO2 measurements?

Pg 9 line 15 an overview the platforms -> an overview about the platforms

Pg 13 line 3 variations of some parameters of a model while others are fixed - isn't this called cross-sensitivity?

Pg 14 line 10 and 12 In a first step -> As a first step

Pg 17 line 11 by COCAP is affected -> by COCAP can be affected

Pg 24 line 15 the wind low -> with low wind speed

Pg 25 Figure 16 : data are the same as in 15 -> data are same as in Figure 15

Relatively low number of references.

---

## Author Comment (AC1) · 30 Jan 2018

**Author's response to Anonymous Referee #1**

We thank the referee for their thoughtful comments on the manuscript. Below, all comments are repeated in italics, followed by our response.

A marked-up version of the revised manuscript (including the changes made in response to both reviews) is attached.

*Pg. 1, line 20: key word is "fully" cover, there are "missed approaches" for research aircraft where they can attempt a landing and get a vertical profile very close to the surface.*

We added the following sentence at p. 1, l. 20: "Missed approaches allow the collection of air samples close to the ground, but this maneuvre may only be performed at sites where the aircraft could actually land."

*The sections on the various types of UAVs seems irrelevant to the paper. Consider saving yourself the cost of ∼1 page and remove or shorten this section to just list them.*

We agree to Referee #1 that UAS are not the focus of this paper and not every reader will be interested in Sect. 2.3. However, we think that the title and structure of this section make it easy to skip if desired. On the other hand, we think that the information given in Sect._2.3 will be valuable for a number of readers as it comprises practical considerations and ballpark figures useful e.g. for project planning. We would therefore prefer to keep the section in the manuscript.

*During the airborne test, interference from turbulence is accounted for. Is there any potential for interference from vibrations from the aircraft (the mirrors in the CO2 sensor perhaps?)? Could you examine this easily and if so, how would one account for this in your instrument design?*

In theory, vibrations and shocks could cause momentary or permanent misalignment of the $CO_2$ sensor's optical assembly. We have no equipment to produce vibrations of defined frequency and amplitude, which would be needed for a systematic analysis. However, the measurements at the Lindenberg observatory were taken on an actual multicopter without special mechanical isolation. Hence, COCAP was subject to increased vibration levels compared to measurements on the ground. As far as we can tell from the data, this has not resulted in increased noise levels or instrument drift. We have added the following to the end of Sect. 4.2.3: "The physical connection between COCAP and the multicopter did not include a dedicated shock absorber (see Fig. S10). Although COCAP's plastic foam housing and the flexibility of the mounting straps provided limited mechanical isolation, sudden movements and vibrations of the multicopter due to turbulence, rotor unbalance and flight manoeuvres have been partially transmitted to the measurement system. In theory, these mechanical disturbances could deteriorate the accuracy of the $x_{CO2}$ measurements, e.g. by causing misalignment of the optical bench of the $CO_2$ sensor. The data collected during the flights at LIN, however, does not exhibit increased noise levels or instrument drift compared to data collected on the ground, suggesting that the movements and vibrations did not degrade COCAP's performance.

We conclude that the measurements gave no indication of (1) calibration problems, (2) uncorrected drift of COCAP between 18 and 27 October, (3) drift of COCAP during

flight or (4) degradation of COCAP's accuracy due to vibrations and sudden movements of the multicopter."

*Sect. 4.2.1: Just to clarify, the drying of the inlet air is still performed on the COCAP, right? The first sentence with "changing humidity" may be a bit confusing, consider changing/rewording.*

We have replaced the first two sentences of Sect. 4.2.1 by: "Testing COCAP under realistic conditions requires the measurement of $x_{CO2}$ in ambient air with varying composition. Specifically, changes in humidity are desirable to reveal potential problems with the drying of the sample gas and changes in $CO_2$ content allow validation of COCAP's calibration curve. To this end, a reference instrument must simultaneously sample the same air as COCAP."

*Pg. 2, line 19: CO2 is defined earlier, but here CH4 is writtten without stating "methane (CH4) earlier (and H2O as well, since it is water vapor specifically). It's generally good practice to define future abbreviations.*

Thank you for bringing this to our attention. As the formulas for water and methane were used in the manuscript only once and twice, respectively, we have replaced them with the spelled out words.

**COCAP: A carbon dioxide analyser for small unmanned aircraft systems**

[revised manuscript text omitted]

The response $x(t)$ of the analyser is the reported $CO_2$ dry air mole fraction. Due to causality, $x(t)$ cannot depend on future $CO_2$ dry air mole fractions at the inlet. Hence, the lower limit of the integration can be set to zero:

$$x(t) = \int_{0}^{\infty} x_{\mathrm{Inlet}}(t - t') \cdot f(t')dt' \tag{13}$$

The instrument function $f(t)$ is not known *a priori*, but can be estimated from the response to a step change in $x_{\mathrm{Inlet}}(t)$. Such step changes occurred at the end of calibration measurements when the supply of gas standard into the sampling line was shut off. From the data we found that for both analysers the response $x_{SC}(t)$ to a step change can be modelled by an exponential decay of the form

$$x_{SC}(t) = (x_0 - x_\infty) \cdot e^{-t/\tau} + x_\infty \tag{14}$$

where $x_0$ and $x_\infty$ are the $CO_2$ dry air mole fractions before and after the step change, respectively, and $\tau$ is the characteristic time constant of the flushing process. We determined time constants of $13\,\mathrm{s}$ for COCAP and $25\,\mathrm{s}$ for the CRDS analyser. To find the function $f(t)$ we differentiate Eq. 13:

$$\frac{d}{dt}x(t) \quad = \quad \frac{d}{dt}\int_0^\infty x_{\text{Inlet}}(t-t') \cdot f(t')dt' \tag{15}$$

$$= \quad \int_0^\infty f(t') \cdot \frac{d}{dt}x_{\text{Inlet}}(t-t')dt' \tag{16}$$

In case of a step change at the inlet, the differentiation yields the Dirac delta function $\delta$, scaled by the height of the step $(x_\infty - x_0)$:

$$\frac{d}{dt}x_{SC}(t) \quad = \quad \int_0^\infty f(t') \cdot (x_\infty - x_0) \cdot \delta(t-t')dt' \tag{17}$$

$$= \quad (x_\infty - x_0) \cdot f(t) \tag{18}$$

Rearranging and applying Eq. 14:

$$f(t) \quad = \quad \frac{\frac{d}{dt}x_{SC}(t)}{x_\infty - x_0} \tag{19}$$

$$= \quad \frac{(x_\infty - x_0) \cdot e^{-t/\tau}}{(x_\infty - x_0) \cdot \tau} \tag{20}$$

$$= \quad \frac{e^{-t/\tau}}{\tau} \tag{21}$$

This means that the instrument function of either analyser can be described with an exponential decay which has the same time constant as the analyser's step response. For practical reasons, we treat $f(t)$ as equal to zero outside $0 \leq t \leq 5\tau$. Because no carbon dioxide is created or removed inside the analysers, the time integral over $f(t)$ must be equal to one, necessitating normalisation of the truncated response function:

$$f'(t) = \begin{cases} 0 & \text{if } t < 0 \\ \frac{f(t)}{1-e^{-5}} & \text{if } 0 \leq t \leq 5\tau \\ 0 & \text{if } t > 5\tau \end{cases} \tag{22}$$

Through the measurement process both analysers effectively convolute the $x_{\text{CO2}}$ signal present at the inlet of the sampling line with their respective instrument function. To make the results comparable, we convolute the measurements of each analyser with the instrument function of *the other* analyser. If both measurements were perfect, this would yield identical results because convolution is commutative:

[Figure]

**Figure 12.** (a) CO₂ dry air mole fraction $x_{CO2}$ of ambient air measured by COCAP and a CRDS analyser during a car drive. The three step-like patterns originate from the field calibration during which two gas standards are sampled. The peak at 9:16 occurred while waiting at a traffic light on a busy street. (b) Difference and smoothed difference of $x_{CO2}$ measured by COCAP minus $x_{CO2}$ measured by the CRDS analyser. The smoothing is implemented by convolution with a Gauss window of 200 s full width at half maximum. The measurements of both analysers have been corrected using the field calibrations and the flushing times have been equalised as explained in the text.

$$x_{COCAP} * f_{CRDS} \quad = \quad (x_{Inlet} * f_{COCAP}) * f_{CRDS} \tag{23}$$

$$= \quad (x_{Inlet} * f_{CRDS}) * f_{COCAP} \tag{24}$$

[revised manuscript text omitted]

The physical connection between COCAP and the multicopter did not include a dedicated shock absorber (see Fig. S10). Although COCAP's plastic foam housing and the flexibility of the mounting straps provided limited mechanical isolation, sudden movements and vibrations of the multicopter due to turbulence, rotor unbalance and flight manoeuvres have been
15    partially transmitted to the measurement system. In theory, these mechanical disturbances could deteriorate the accuracy of the $x_{CO2}$ measurements, e.g. by causing misalignment of the optical bench of the $CO_2$ sensor. The data collected during the flights at LIN, however, does not exhibit increased noise levels or instrument drift compared to data collected on the ground, suggesting that the movements and vibrations did not degrade COCAP's performance.

**Table 3.** Mean of the difference between COCAP's $CO_2$ dry air mole fraction measurements $x_{COCAP}$ and the corresponding measurements by LIN $x_{LIN}$ ($\pm$ 1 standard error). Subsets of COCAP's measurements are also analysed.

| COCAP measurements | $\overline{x_{COCAP} - x_{LIN}}$ in µmol·mol$^{-1}$ |
|---|---|
| All | $0.23 \pm 0.45$ |
| 18 October | $0.16 \pm 0.85$ |
| 27 October | $0.28 \pm 0.49$ |
| All first ascents | $0.39 \pm 0.66$ |
| All second ascents | $0.06 \pm 0.52$ |

**Table 4.** Statistical tests for bias. Here $x$ denotes measurements by COCAP and the index defines a subset: 'A' for all measurements, '18' and '27' for 18 and 27 October, respectively, and 'A1' and 'A2' for first and second ascent, respectively. $D(x)$ represents the difference between $x$ and the corresponding measurements by LIN. An overline denotes the arithmetic mean.

| Null hypothesis | Statistical test | Test result $p$ |
|---|---|---|
| $\overline{D(x_C)} = 0$ | Welch's $t$-test | 0.75 |
| $\overline{D(x_{18})} = \overline{D(x_{27})}$ | Welch's $t$-test | 0.72 |
| $\overline{D(x_{A1})} = \overline{D(x_{A2})}$ | Student's $t$-test | 0.75 |

We conclude that the measurements gave no indication of (1) calibration problems, (2) uncorrected drift of COCAP between 18 and 27 October  (3) drift of COCAP during flight or (4) degradation of COCAP's accuracy due to vibrations and sudden movements of the multicopter.

**5 Summary and conclusions**

With COCAP we have designed and built a self-contained analyser for the measurement of $CO_2$ dry air mole fraction, temperature, humidity and pressure of ambient air on board UAS. COCAP is typically operated under ambient conditions that change quickly and over wide ranges. These challenging conditions can compromise the accuracy of $CO_2$ sensors. We ensure COCAP's accuracy by (1) temperature stabilisation, (2) drying of sample air, (3) a calibration curve that includes correction terms for temperature and pressure and (4) by regular field calibrations. When high-frequency noise is filtered out, COCAP's measurements of $CO_2$ dry air mole fraction were found to deviate from a reference by not more than 1.2 µmol·mol$^{-1}$ during simulated flights and by not more than 1 µmol·mol$^{-1}$ during deployment in an instrumented van. In a comparison to the ICOS observatory Lindenberg no indication of bias or uncorrected drift was observed.

Since the design of COCAP, newer versions of SenseAir's HPP sensor family have become available. They exhibit lower drift and lower noise at a slightly smaller form factor (Arzoumanian et al., 2016). The integration of the newer sensors into COCAP would be straightforward and is expected to further improve the accuracy of the $x_{CO2}$ measurements.

With a volume of $14 \times 14 \times 42 \, \text{cm}^3$ and a mass of $1 \, \text{kg}$ COCAP fits onto small UAS with a take-off mass below $5 \, \text{kg}$. It is therefore a cost-effective tool to study carbon dioxide in the lowest $100$–$1000 \, \text{m}$ of Earth's atmosphere. On a multicopter or fixed-wing aircraft COCAP enables measurements at a finer scale than manned aircraft and without restrictions of minimum flight altitude. On a tethered balloon, COCAP can take measurements for longer time spans without being bound to fixed altitudes like an instrumented mast or tower.

The techniques presented in this article are applicable to other measurement systems as well. Many sensors benefit from a stable temperature and we have shown how an effective temperature stabilisation can be achieved within the mass, size and power restrictions of a small UAS. Likewise, the presented method for obtaining a calibration curve can be applied to other gas sensors. Regular calibrations are important to ensure the accuracy of trace gas measurements and we have given an example how to implement them in a practicable way.

Flying a $CO_2$ analyser on small UAS opens up new possibilities in studying the carbon cycle. As a first application we have constrained nocturnal carbon dioxide fluxes of vegetation using series of $x_{CO2}$ profiles in a budget method (Kunz et al., in preparation). A similar approach could be used to estimate $CO_2$ emissions of cities, ideally by simultaneously deploying several UAS at different downwind locations. Furthermore, the strength of point sources like power plants or factories could be estimated by applying a mass balance technique as is commonly used in aircraft-based studies (Conley et al., 2017 and references therein). The main advantages of small UAS over manned aircraft in these applications is their full vertical coverage from the ground to several hundred meters height and their much lower acquisition and operating cost. As small unmanned aircraft are typically limited to a range between 1 and 50 km in a single flight, they are best suited for studying smaller areas. Their low air speed and high manoeuvrability enables them to sample the atmosphere with high spatial resolution. Equipped with an analyser for carbon dioxide UAS could also become a powerful validation tool for efforts to model dispersion of tracers on fine scales, e.g. inside street canyons.

Due to their unique capabilities and low cost, we foresee that the use of unmanned aircraft in the Earth sciences will significantly increase in the near future. We have shown how accurate measurements of the $CO_2$ dry air mole fraction can be taken on board small UAS and we anticipate these platforms to play an important role in closing gaps in the observation of the carbon cycle.

*Data availability.* The analyses presented here are based on many different experiments and use a combination of two or three different data sources in most cases. Compiling the data into a uniform, self-describing collection suitable for upload to a repository would be a great effort. Given the fact that our experiments were aimed at characterising COCAP, reuse of the data by other groups seems unlikely. Hence, we did not upload our measurement data to a repository. However, data from individual experiments is available from the corresponding author upon request.

Data from the ICOS station Lindenberg can be requested from ICOS-D (http://www.icos-infrastruktur.de/en/mitarbeiter/atmosphaerenprogramm/).

*Competing interests.* Christine Hummelgård, Hans Martin and Henrik Rödjegård work for SenseAir AB, the manufacturer of the HPP sensor family. The other authors declare that they have no conflict of interest.

*Acknowledgements.* We thank Maksym Bryzgalov (SenseAir AB, Sweden) for helping with the configuration and integration of the $CO_2$ sensor. Hök Instruments AB (Sweden) kindly provided software for COCAP's data logger. We thank Wieland Jeschag and Till Fastnacht for adapting this software to our needs. We acknowledge Jürgen Kaulfuß, who designed and built the field calibration device. Calibration standards were prepared by the gas lab of the MPI for Biogeochemistry Jena and we are grateful for their support. We thank Frank Beyrich, Matthias Lindauer, Udo Rummel and Marcus Schumacher (Deutscher Wetterdienst, Germany) for access to the Lindenberg station, technical support and data sharing. We gratefully acknowledge the authors of various open-source software packages that were used in the project and for the preparation of the manuscript, in particular GNU Octave (including the *optim* package), KiCad, gnuplot, GIMP, Inkscape, LyX and LaTeX. We thank the Max Planck Society for generous financial support. Parts ofM.K. this work have been funded byks COST (European cooperation in science and technology) and the German academic exchange service DAAD for funding.

---

## Author Comment (AC2) · 30 Jan 2018

**Comments from Anonymous Referee #2**

We thank the referee for their thoughtful comments on the manuscript. Below, all comments are repeated in italics, followed by our response.

A marked-up version of the revised manuscript (including the changes made in response to both reviews) is attached.

*It would be interesting to see in the conclusion more future directions that focus on different scientific questions that could be answered with this measurement method. Based on the paper the main purpose of this measurement method is "close gaps in the observation of the carbon cycle", however maybe try expanding this statement. You refer to this in the introduction and conclusion but maybe highlight more what is and isn't possible to do with this measurement method. Explain more which gaps you are trying to close and why.*

*Similarly in the conclusion line 25 you state 'Other potential uses include measuring the emission strength of point sources and investigation of emissions in urban areas', yes this is possible with your measurement method but it is also possible with numerous other instruments. It does not emphasise a unique benefit of this instrument. Also you wouldn't actually measure the emission strength directly, but make measurements that can be used to investigate the strength of point sources.*

We have expanded the outlook. What was formerly p. 26 ll. 28 now reads: "As a first application we have constrained nocturnal carbon dioxide fluxes of vegetation using series of $x_{CO2}$ profiles in a budget method (Kunz et al., in preparation). A similar approach could be used to estimate $CO_2$ emissions of cities, ideally by simultaneously deploying several UAS at different downwind locations. Furthermore, the strength of point sources like power plants or factories could be estimated by applying a mass balance technique as is commonly used in aircraft-based studies (Conley et al., 2017 and references therein). The main advantages of small UAS over manned aircraft in these applications is their full vertical coverage from the ground to several hundred meters height and their much lower acquisition and operating cost. As small unmanned aircraft are typically limited to a range between 1 and 50 km in a single flight, they are best suited for studying smaller areas. Their low air speed and high manoeuvrability enables them to sample the atmosphere with high spatial resolution. Equipped with an analyser for carbon dioxide UAS could also become a powerful validation tool for efforts to model dispersion of tracers on fine scales, e.g. inside street canyons."

*It would be good to add a table with the approximate height, area, time that can be covered when they use the UAS (with details about the specific setup and when they take into account everything e.g. battery life etc) relative to other platforms described in the paper.*

We agree to Referee #1 that UAS are not the focus of this paper. Therefore, we are hesitant to add additional technical details regarding the platforms.

*Change the x and y labels on the plots - Time in h change to Time (h) or better to Time / h*

Done.

*Section 2.1.5 is there a reference for all the 5 statements of how temperature affects CO2 measurements?*

We have added references for temperature effects on our measurements.

*Pg 9 line 15 an overview the platforms -> an overview about the platforms*

This refers to the version of the manuscript originally submitted. The respective sentence was removed following comments during Quick Access Review.

*Pg 13 line 3 variations of some parameters of a model while others are fixed - isn't this called cross-sensitivity?*

That paragraph is about the fitting procedure, not about properties of the parameters. We used a gradient method to optimise a function in a high-dimensional parameter space, which generally involves the risk of divergence or convergence to a local optimum that is not the global optimum. To avoid such problems, we started the multi-step fitting by varying few very important parameters. The less influential, higher-order terms were kept constant and only included in later steps of the fitting. Another way to look at this is that we fitted a simple model in the beginning and successively refined it by replacing constants with parameters that were also varied during the optimisation. The capabilities of the *leasqr* function made this particularly easy. We think that the paragraph would benefit from including the term "cross-sensitivity".

*Pg 14 line 10 and 12 In a first step -> As a first step*

Changed

*Pg 17 line 11 by COCAP is affected -> by COCAP can be affected*

As none of the error sources listed in this sentence can be fully eliminated, we think that the current wording is appropriate.

*Pg 24 line 15 the wind low -> with low wind speed*

Changed to "[...] the wind speed was low [...]"

*Pg 25 Figure 16 : data are the same as in 15 -> data are same as in Figure 15*

Changed.

*Relatively low number of references.*

We have added references to Section 2.1.5 as suggested (see above).

**COCAP: A carbon dioxide analyser for small unmanned aircraft systems**

[revised manuscript text omitted]

The response $x(t)$ of the analyser is the reported $CO_2$ dry air mole fraction. Due to causality, $x(t)$ cannot depend on future $CO_2$ dry air mole fractions at the inlet. Hence, the lower limit of the integration can be set to zero:

$$x(t) = \int_{0}^{\infty} x_{\mathrm{Inlet}}(t - t') \cdot f(t')dt' \tag{13}$$

The instrument function $f(t)$ is not known *a priori*, but can be estimated from the response to a step change in $x_{\mathrm{Inlet}}(t)$. Such step changes occurred at the end of calibration measurements when the supply of gas standard into the sampling line was shut off. From the data we found that for both analysers the response $x_{SC}(t)$ to a step change can be modelled by an exponential decay of the form

$$x_{SC}(t) = (x_0 - x_\infty) \cdot e^{-t/\tau} + x_\infty \tag{14}$$

where $x_0$ and $x_\infty$ are the $CO_2$ dry air mole fractions before and after the step change, respectively, and $\tau$ is the characteristic time constant of the flushing process. We determined time constants of $13\,s$ for COCAP and $25\,s$ for the CRDS analyser. To find the function $f(t)$ we differentiate Eq. 13:

$$\frac{d}{dt}x(t) = \frac{d}{dt}\int_0^\infty x_{\text{Inlet}}(t-t') \cdot f(t')dt' \tag{15}$$

$$= \int_0^\infty f(t') \cdot \frac{d}{dt}x_{\text{Inlet}}(t-t')dt' \tag{16}$$

In case of a step change at the inlet, the differentiation yields the Dirac delta function $\delta$, scaled by the height of the step $(x_\infty - x_0)$:

$$\frac{d}{dt}x_{SC}(t) = \int_0^\infty f(t') \cdot (x_\infty - x_0) \cdot \delta(t-t')dt' \tag{17}$$

$$= (x_\infty - x_0) \cdot f(t) \tag{18}$$

Rearranging and applying Eq. 14:

$$f(t) = \frac{\frac{d}{dt}x_{SC}(t)}{x_\infty - x_0} \tag{19}$$

$$= \frac{(x_\infty - x_0) \cdot e^{-t/\tau}}{(x_\infty - x_0) \cdot \tau} \tag{20}$$

$$= \frac{e^{-t/\tau}}{\tau} \tag{21}$$

This means that the instrument function of either analyser can be described with an exponential decay which has the same time constant as the analyser's step response. For practical reasons, we treat $f(t)$ as equal to zero outside $0 \leq t \leq 5\tau$. Because no carbon dioxide is created or removed inside the analysers, the time integral over $f(t)$ must be equal to one, necessitating normalisation of the truncated response function:

$$f'(t) = \begin{cases} 0 & \text{if } t < 0 \\ \frac{f(t)}{1-e^{-5}} & \text{if } 0 \leq t \leq 5\tau \\ 0 & \text{if } t > 5\tau \end{cases} \tag{22}$$

Through the measurement process both analysers effectively convolute the $x_{\text{CO2}}$ signal present at the inlet of the sampling line with their respective instrument function. To make the results comparable, we convolute the measurements of each analyser with the instrument function of *the other* analyser. If both measurements were perfect, this would yield identical results because convolution is commutative:

[Figure]

**Figure 12.** (a) $CO_2$ dry air mole fraction $x_{CO2}$ of ambient air measured by COCAP and a CRDS analyser during a car drive. The three step-like patterns originate from the field calibration during which two gas standards are sampled. The peak at 9:16 occurred while waiting at a traffic light on a busy street. (b) Difference and smoothed difference of $x_{CO2}$ measured by COCAP minus $x_{CO2}$ measured by the CRDS analyser. The smoothing is implemented by convolution with a Gauss window of 200 s full width at half maximum. The measurements of both analysers have been corrected using the field calibrations and the flushing times have been equalised as explained in the text.

$$x_{COCAP} * f_{CRDS} \quad = \quad (x_{Inlet} * f_{COCAP}) * f_{CRDS} \tag{23}$$

$$= \quad (x_{Inlet} * f_{CRDS}) * f_{COCAP} \tag{24}$$

[revised manuscript text omitted]

The physical connection between COCAP and the multicopter did not include a dedicated shock absorber (see Fig. S10). Although COCAP's plastic foam housing and the flexibility of the mounting straps provided limited mechanical isolation, sudden movements and vibrations of the multicopter due to turbulence, rotor unbalance and flight manoeuvres have been
15  partially transmitted to the measurement system. In theory, these mechanical disturbances could deteriorate the accuracy of the $x_{CO_2}$ measurements, e.g. by causing misalignment of the optical bench of the $CO_2$ sensor. The data collected during the flights at LIN, however, does not exhibit increased noise levels or instrument drift compared to data collected on the ground, suggesting that the movements and vibrations did not degrade COCAP's performance.

**Table 3.** Mean of the difference between COCAP's $CO_2$ dry air mole fraction measurements $x_{COCAP}$ and the corresponding measurements by LIN $x_{LIN}$ ($\pm$ 1 standard error). Subsets of COCAP's measurements are also analysed.

| COCAP measurements | $\overline{x_{COCAP} - x_{LIN}}$ in $\mu mol \cdot mol^{-1}$ |
|---|---|
| All | $0.23 \pm 0.45$ |
| 18 October | $0.16 \pm 0.85$ |
| 27 October | $0.28 \pm 0.49$ |
| All first ascents | $0.39 \pm 0.66$ |
| All second ascents | $0.06 \pm 0.52$ |

**Table 4.** Statistical tests for bias. Here $x$ denotes measurements by COCAP and the index defines a subset: 'A' for all measurements, '18' and '27' for 18 and 27 October, respectively, and 'A1' and 'A2' for first and second ascent, respectively. $D(x)$ represents the difference between $x$ and the corresponding measurements by LIN. An overline denotes the arithmetic mean.

| Null hypothesis | Statistical test | Test result $p$ |
|---|---|---|
| $\overline{D(x_C)} = 0$ | Welch's $t$-test | 0.75 |
| $\overline{D(x_{18})} = \overline{D(x_{27})}$ | Welch's $t$-test | 0.72 |
| $\overline{D(x_{A1})} = \overline{D(x_{A2})}$ | Student's $t$-test | 0.75 |

We conclude that the measurements gave no indication of (1) calibration problems, (2) uncorrected drift of COCAP between 18 and 27 October or, (3) drift of COCAP during flight or (4) degradation of COCAP's accuracy due to vibrations and sudden movements of the multicopter.

**5 Summary and conclusions**

With COCAP we have designed and built a self-contained analyser for the measurement of $CO_2$ dry air mole fraction, temperature, humidity and pressure of ambient air on board UAS. COCAP is typically operated under ambient conditions that change quickly and over wide ranges. These challenging conditions can compromise the accuracy of $CO_2$ sensors. We ensure COCAP's accuracy by (1) temperature stabilisation, (2) drying of sample air, (3) a calibration curve that includes correction terms for temperature and pressure and (4) by regular field calibrations. When high-frequency noise is filtered out, COCAP's measurements of $CO_2$ dry air mole fraction were found to deviate from a reference by not more than $1.2\,\mu mol \cdot mol^{-1}$ during simulated flights and by not more than $1\,\mu mol \cdot mol^{-1}$ during deployment in an instrumented van. In a comparison to the ICOS observatory Lindenberg no indication of bias or uncorrected drift was observed.

Since the design of COCAP, newer versions of SenseAir's HPP sensor family have become available. They exhibit lower drift and lower noise at a slightly smaller form factor (Arzoumanian et al., 2016). The integration of the newer sensors into COCAP would be straightforward and is expected to further improve the accuracy of the $x_{CO2}$ measurements.

With a volume of $14 \times 14 \times 42 \, \text{cm}^3$ and a mass of $1 \, \text{kg}$ COCAP fits onto small UAS with a take-off mass below $5 \, \text{kg}$. It is therefore a cost-effective tool to study carbon dioxide in the lowest 100–1000 m of Earth's atmosphere. On a multicopter or fixed-wing aircraft COCAP enables measurements at a finer scale than manned aircraft and without restrictions of minimum flight altitude. On a tethered balloon, COCAP can take measurements for longer time spans without being bound to fixed altitudes like an instrumented mast or tower.

The techniques presented in this article are applicable to other measurement systems as well. Many sensors benefit from a stable temperature and we have shown how an effective temperature stabilisation can be achieved within the mass, size and power restrictions of a small UAS. Likewise, the presented method for obtaining a calibration curve can be applied to other gas sensors. Regular calibrations are important to ensure the accuracy of trace gas measurements and we have given an example how to implement them in a practicable way.

Flying a $CO_2$ analyser on small UAS opens up new possibilities in studying the carbon cycle. As a first application we have constrained nocturnal carbon dioxide fluxes of vegetation using seriess of $x_{CO2}$ profiles in a budget method (Kunz et al., in preparation). A similar approach could be used to estimate $CO_2$ emissions of cities, ideally by simultaneously deploying several UAS at different downwind locations. Furthermore, the strength of point sources like power plants or factories could be estimated by applying a mass balance technique as is commonly used in aircraft-based studies (Conley et al., 2017 and references therein). The main advantages of small UAS over manned aircraft in these applications is their full vertical coverage from the ground to several hundred meters height and their much lower acquisition and operating cost. As small unmanned aircraft are typically limited to a range between 1 and 50 km in a single flight, they are best suited for studying smaller areas. Their low air speed and high manoeuvrability enables them to sample the atmosphere with high spatial resolution. Equipped with an analyser for carbon dioxide UAS could also become a powerful validation tool for efforts to model dispersion of tracers on fine scales, e.g. inside street canyons.

Due to their unique capabilities and low cost, we foresee that the use of unmanned aircraft in the Earth sciences will significantly increase in the near future. We have shown how accurate measurements of the $CO_2$ dry air mole fraction can be taken on board small UAS and we anticipate these platforms to play an important role in closing gaps in the observation of the carbon cycle.

*Data availability.* The analyses presented here are based on many different experiments and use a combination of two or three different data sources in most cases. Compiling the data into a uniform, self-describing collection suitable for upload to a repository would be a great effort. Given the fact that our experiments were aimed at characterising COCAP, reuse of the data by other groups seems unlikely. Hence, we did not upload our measurement data to a repository. However, data from individual experiments is available from the corresponding author upon request.

Data from the ICOS station Lindenberg can be requested from ICOS-D (http://www.icos-infrastruktur.de/en/mitarbeiter/atmosphaerenprogramm/).

*Competing interests.* Christine Hummelgård, Hans Martin and Henrik Rödjegård work for SenseAir AB, the manufacturer of the HPP sensor family. The other authors declare that they have no conflict of interest.

*Acknowledgements.* We thank Maksym Bryzgalov (SenseAir AB, Sweden) for helping with the configuration and integration of the $CO_2$ sensor. Hök Instruments AB (Sweden) kindly provided software for COCAP's data logger. We thank Wieland Jeschag and Till Fastnacht for adapting this software to our needs. We acknowledge Jürgen Kaulfuß, who designed and built the field calibration device. Calibration standards were prepared by the gas lab of the MPI for Biogeochemistry Jena and we are grateful for their support. We thank Frank Beyrich, Matthias Lindauer, Udo Rummel and Marcus Schumacher (Deutscher Wetterdienst, Germany) for access to the Lindenberg station, technical support and data sharing. We gratefully acknowledge the authors of various open-source software packages that were used in the project and for the preparation of the manuscript, in particular GNU Octave (including the *optim* package), KiCad, gnuplot, GIMP, Inkscape, LyX and LaTeX. We thank the Max Planck Society for generous financial support. Parts ofM.K. this work have been funded byks COST (European cooperation in science and technology) and the German academic exchange service DAAD for funding.

---

## Author Response (AR1)

**Author's response to the referees**

30 January 2018

We thank the two referees for their thoughtful comments on the manuscript. Below, all comments are repeated in italics, followed by our response.

A marked-up version of the revised manuscript is attached.

**Comments from Anonymous Referee #1**

*Pg. 1, line 20: key word is "fully" cover, there are "missed approaches" for research aircraft where they can attempt a landing and get a vertical profile very close to the surface.*

We added the following sentence at p. 1, l. 20: "Missed approaches allow the collection of air samples close to the ground, but this maneuvre may only be performed at sites where the aircraft could actually land."

*The sections on the various types of UAVs seems irrelevant to the paper. Consider saving yourself the cost of ~1 page and remove or shorten this section to just list them.*

We agree to Referee #1 that UAS are not the focus of this paper and not every reader will be interested in Sect. 2.3. However, we think that the title and structure of this section make it easy to skip if desired. On the other hand, we think that the information given in Sect. 2.3 will be valuable for a number of readers as it comprises practical considerations and ballpark figures useful e.g. for project planning. We would therefore prefer to keep the section in the manuscript.

*During the airborne test, interference from turbulence is accounted for. Is there any potential for interference from vibrations from the aircraft (the mirrors in the CO2 sensor perhaps?)? Could you examine this easily and if so, how would one account for this in your instrument design?*

In theory, vibrations and shocks could cause momentary or permanent misalignment of the $CO_2$ sensor's optical assembly. We have no equipment to produce vibrations of defined frequency and amplitude, which would be needed for a systematic analysis. However, the measurements at the Lindenberg observatory were taken on an actual multicopter without special mechanical isolation. Hence, COCAP was subject to increased vibration levels compared to measurements on the ground. As far as we can tell from the data, this has not resulted in increased noise levels or instrument drift. We have added the following to the end of Sect. 4.2.3: "The physical connection between COCAP and the multicopter

did not include a dedicated shock absorber (see Fig. S10). Although COCAP's plastic foam housing and the flexibility of the mounting straps provided limited mechanical isolation, sudden movements and vibrations of the multicopter due to turbulence, rotor unbalance and flight manoeuvres have been partially transmitted to the measurement system. In theory, these mechanical disturbances could deteriorate the accuracy of the $x_{CO2}$ measurements, e.g. by causing misalignment of the optical bench of the $CO_2$ sensor. The data collected during the flights at LIN, however, does not exhibit increased noise levels or instrument drift compared to data collected on the ground, suggesting that the movements and vibrations did not degrade COCAP's performance.

We conclude that the measurements gave no indication of (1) calibration problems, (2) uncorrected drift of COCAP between 18 and 27 October, (3) drift of COCAP during flight or (4) degradation of COCAP's accuracy due to vibrations and sudden movements of the multicopter."

*Sect. 4.2.1: Just to clarify, the drying of the inlet air is still performed on the COCAP, right? The first sentence with "changing humidity" may be a bit confusing, consider changing/rewording.*

We have replaced the first two sentences of Sect. 4.2.1 by: "Testing COCAP under realistic conditions requires the measurement of $x_{CO2}$ in ambient air with varying composition. Specifically, changes in humidity are desirable to reveal potential problems with the drying of the sample gas and changes in $CO_2$ content allow validation of COCAP's calibration curve. To this end, a reference instrument must simultaneously sample the same air as COCAP."

*Pg. 2, line 19: CO2 is defined earlier, but here CH4 is writtten without stating "methane (CH4) earlier (and H2O as well, since it is water vapor specifically). It's generally good practice to define future abbreviations.*

Thank you for bringing this to our attention. As the formulas for water and methane were used in the manuscript only once and twice, respectively, we have replaced them with the spelled out words.

**Comments from Anonymous Referee #2**

*It would be interesting to see in the conclusion more future directions that focus on different scientific questions that could be answered with this measurement method. Based on the paper the main purpose of this measurement method is "close gaps in the observation of the carbon cycle", however maybe try expanding this statement. You refer to this in the introduction and conclusion but maybe highlight more what is and isn't possible to do with this measurement method. Explain more which gaps you are trying to close and why.*

*Similarly in the conclusion line 25 you state 'Other potential uses include measuring the emission strength of point sources and investigation of emissions in urban areas', yes this is possible with your measurement method but it is also possible with numerous other instruments. It does not emphasise a unique benefit of this instrument. Also you wouldn't actually measure the emission strength directly, but make measurements that can be used*

*to investigate the strength of point sources.*

We have expanded the outlook. What was formerly p. 26 ll. 28 now reads: "As a first application we have constrained nocturnal carbon dioxide fluxes of vegetation using series of $x_{CO_2}$ profiles in a budget method (Kunz et al., in preparation). A similar approach could be used to estimate $CO_2$ emissions of cities, ideally by simultaneously deploying several UAS at different downwind locations. Furthermore, the strength of point sources like power plants or factories could be estimated by applying a mass balance technique as is commonly used in aircraft-based studies (Conley et al., 2017 and references therein). The main advantages of small UAS over manned aircraft in these applications is their full vertical coverage from the ground to several hundred meters height and their much lower acquisition and operating cost. As small unmanned aircraft are typically limited to a range between 1 and 50 km in a single flight, they are best suited for studying smaller areas. Their low air speed and high manoeuvrability enables them to sample the atmosphere with high spatial resolution. Equipped with an analyser for carbon dioxide UAS could also become a powerful validation tool for efforts to model dispersion of tracers on fine scales, e.g. inside street canyons."

*It would be good to add a table with the approximate height, area, time that can be covered when they use the UAS (with details about the specific setup and when they take into account everything e.g. battery life etc) relative to other platforms described in the paper.*

We agree to Referee #1 that UAS are not the focus of this paper. Therefore, we are hesitant to add additional technical details regarding the platforms.

*Change the x and y labels on the plots - Time in h change to Time (h) or better to Time / h*

Done.

*Section 2.1.5 is there a reference for all the 5 statements of how temperature affects CO2 measurements?*

We have added references for temperature effects on our measurements.

*Pg 9 line 15 an overview the platforms -> an overview about the platforms*

This refers to the version of the manuscript originally submitted. The respective sentence was removed following comments during Quick Access Review.

*Pg 13 line 3 variations of some parameters of a model while others are fixed - isn't this called cross-sensitivity?*

That paragraph is about the fitting procedure, not about properties of the parameters. We used a gradient method to optimise a function in a high-dimensional parameter space, which generally involves the risk of divergence or convergence to a local optimum that is not the global optimum. To avoid such problems, we started the multi-step fitting by varying few very important parameters. The less influential, higher-order terms were kept constant and only included in later steps of the fitting. Another way to look at this is that we fitted a simple model in the beginning and successively refined it by replacing constants with parameters that were also varied during the optimisation. The capabilities of the *leasqr* function made this particularly easy. We think that the paragraph would benefit from including the term "cross-sensitivity".

*Pg 14 line 10 and 12 In a first step -> As a first step*

Changed

*Pg 17 line 11 by COCAP is affected -> by COCAP can be affected*

As none of the error sources listed in this sentence can be fully eliminated, we think that the current wording is appropriate.

*Pg 24 line 15 the wind low -> with low wind speed*

Changed to "[...] the wind speed was low [...]"

*Pg 25 Figure 16 : data are the same as in 15 -> data are same as in Figure 15*

Changed.

*Relatively low number of references.*

We have added references to Section 2.1.5 as suggested (see above).

**COCAP: A carbon dioxide analyser for small unmanned aircraft systems**

Martin Kunz[1], Jost V. Lavric[1], Christoph Gerbig[1], Pieter Tans[2], Don Neff[2], Christine Hummelgård[3], Hans Martin[3], Henrik Rödjegård[3], Burkhard Wrenger[4], and Martin Heimann[1,5]

[1]Max Planck Institute for Biogeochemistry, Jena, Germany
[2]NOAA Earth System Research Laboratory, Global Monitoring Division, Boulder, Colorado, USA
[3]SenseAir AB, Delsbo, Sweden
[4]Ostwestfalen-Lippe University of Applied Sciences, Hoexter, Germany
[5]Division of Atmospheric Sciences, Department of Physics, University of Helsinki, Finland

*Correspondence to:* Martin Kunz (mkunz@bgc-jena.mpg.de)

**Abstract.** Unmanned aircraft systems (UAS) could provide a cost-effective way to close gaps in the observation of the carbon cycle, provided that small yet accurate analysers are available. We have developed a COmpact Carbon dioxide analyser for Airborne Platforms (COCAP). The accuracy of COCAP's carbon dioxide ($CO_2$) measurements is ensured by calibration in an environmental chamber, regular calibration in the field and by chemical drying of sampled air. In addition, the package contains a lightweight thermal stabilisation system that reduces the influence of ambient temperature changes on the $CO_2$ sensor by two orders of magnitude. During validation of COCAP's $CO_2$ measurements in simulated and real flights we found a measurement error of $1.2\,\mu mol \cdot mol^{-1}$ or better with no indication of bias. COCAP is a self-contained package that has proven well suited for the operation on board small UAS. Besides carbon dioxide dry air mole fraction it also measures air temperature, humidity and pressure. We describe the measurement system and our calibration strategy in detail to support others in tapping the potential of UAS for atmospheric trace gas measurements.

**1 Introduction**

Atmospheric measurements of carbon dioxide ($CO_2$) are essential for our understanding of the carbon cycle and how it changes in a warming climate. Such measurements are made on a regular basis by global networks of surface stations, by specially instrumented aircraft and by research ships (Masarie and Tans, 1995). When local influences are filtered out, the data from these measurements allow the identification of global trends and the characterisation of major greenhouse gas sources and sinks, generally on the scale of continents (Fan et al., 1998; Ciais et al., 2010). This top-down approach to the quantification of the carbon cycle is well established.

In contrast, for observations on smaller scales conventional strategies often suffer from severe limitations. Specifically, the transition region between micro- and mesoscale in the sense of Orlanski (1975) poses a challenge. It comprises horizontal extents of 200 m to 20 km and periods from minutes to hours. Manned research aircraft do not fully cover this region due to minimum flight altitude requirements and their in most cases high airspeed. Missed approaches allow the collection of

air samples close to the ground, but this manoeuvre may only be performed at sites where the aircraft could actually land. Furthermore, because  operation of manned aircraft is costly, 
[revised manuscript text omitted]

The physical connection between COCAP and the multicopter did not include a dedicated shock absorber (see Fig. S10). Although COCAP's plastic foam housing and the flexibility of the mounting straps provided limited mechanical isolation, sudden movements and vibrations of the multicopter due to turbulence, rotor unbalance and flight manoeuvres have been
15   partially transmitted to the measurement system. In theory, these mechanical disturbances could deteriorate the accuracy of the $x_{CO2}$ measurements, e.g. by causing misalignment of the optical bench of the $CO_2$ sensor. The data collected during the flights at LIN, however, does not exhibit increased noise levels or instrument drift compared to data collected on the ground, suggesting that the movements and vibrations did not degrade COCAP's performance.

**Table 3.** Mean of the difference between COCAP's $CO_2$ dry air mole fraction measurements $x_{\text{COCAP}}$ and the corresponding measurements by LIN $x_{\text{LIN}}$ ($\pm$ 1 standard error). Subsets of COCAP's measurements are also analysed.

| COCAP measurements | $\overline{x_{\text{COCAP}} - x_{\text{LIN}}}$ in µmol·mol$^{-1}$ |
|---|---|
| All | $0.23 \pm 0.45$ |
| 18 October | $0.16 \pm 0.85$ |
| 27 October | $0.28 \pm 0.49$ |
| All first ascents | $0.39 \pm 0.66$ |
| All second ascents | $0.06 \pm 0.52$ |

**Table 4.** Statistical tests for bias. Here $x$ denotes measurements by COCAP and the index defines a subset: 'A' for all measurements, '18' and '27' for 18 and 27 October, respectively, and 'A1' and 'A2' for first and second ascent, respectively. $D(x)$ represents the difference between $x$ and the corresponding measurements by LIN. An overline denotes the arithmetic mean.

| Null hypothesis | Statistical test | Test result $p$ |
|---|---|---|
| $\overline{D(x_{\text{C}})} = 0$ | Welch's $t$-test | 0.75 |
| $\overline{D(x_{18})} = \overline{D(x_{27})}$ | Welch's $t$-test | 0.72 |
| $\overline{D(x_{\text{A1}})} = \overline{D(x_{\text{A2}})}$ | Student's $t$-test | 0.75 |

We conclude that the measurements gave no indication of (1) calibration problems, (2) uncorrected drift of COCAP between 18 and 27 October or, (3) drift of COCAP during flight or (4) degradation of COCAP's accuracy due to vibrations and sudden movements of the multicopter.

**5 Summary and conclusions**

With COCAP we have designed and built a self-contained analyser for the measurement of $CO_2$ dry air mole fraction, temperature, humidity and pressure of ambient air on board UAS. COCAP is typically operated under ambient conditions that change quickly and over wide ranges. These challenging conditions can compromise the accuracy of $CO_2$ sensors. We ensure COCAP's accuracy by (1) temperature stabilisation, (2) drying of sample air, (3) a calibration curve that includes correction terms for temperature and pressure and (4) by regular field calibrations. When high-frequency noise is filtered out, COCAP's measurements of $CO_2$ dry air mole fraction were found to deviate from a reference by not more than 1.2 µmol·mol$^{-1}$ during simulated flights and by not more than 1 µmol·mol$^{-1}$ during deployment in an instrumented van. In a comparison to the ICOS observatory Lindenberg no indication of bias or uncorrected drift was observed.

Since the design of COCAP, newer versions of SenseAir's HPP sensor family have become available. They exhibit lower drift and lower noise at a slightly smaller form factor (Arzoumanian et al., 2016). The integration of the newer sensors into COCAP would be straightforward and is expected to further improve the accuracy of the $x_{CO2}$ measurements.

With a volume of $14{\times}14{\times}42\,\mathrm{cm}^3$and a mass of $1\,\mathrm{kg}$ COCAP fits onto small UAS with a take-off mass below $5\,\mathrm{kg}$. It is therefore a cost-effective tool to study carbon dioxide in the lowest $100$–$1000\,\mathrm{m}$ of Earth's atmosphere. On a multicopter or fixed-wing aircraft COCAP enables measurements at a finer scale than manned aircraft and without restrictions of minimum flight altitude. On a tethered balloon, COCAP can take measurements for longer time spans without being bound to fixed altitudes like an instrumented mast or tower.

The techniques presented in this article are applicable to other measurement systems as well. Many sensors benefit from a stable temperature and we have shown how an effective temperature stabilisation can be achieved within the mass, size and power restrictions of a small UAS. Likewise, the presented method for obtaining a calibration curve can be applied to other gas sensors. Regular calibrations are important to ensure the accuracy of trace gas measurements and we have given an example how to implement them in a practicable way.

Flying a $CO_2$ analyser on small UAS opens up new possibilities in studying the carbon cycle. As a first application we have constrained nocturnal carbon dioxide fluxes of vegetation using repeated series of $x_{CO2}$ profiles in a budget method (Kunz et al., in preparation). A similar approach could be used to estimate $CO_2$ emissions of cities, ideally by simultaneously deploying several UAS at different downwind locations. Furthermore, the strength of point sources like power plants or factories could be estimated by applying a mass balance technique as is commonly used in aircraft-based studies (Conley et al., 2017 and references therein). The main advantages of small UAS over manned aircraft in these applications is their full vertical coverage from the ground to several hundred meters height and their much lower acquisition and operating cost. As small unmanned aircraft are typically limited to a range between 1 and $50\,\mathrm{km}$ in a single flight, they are best suited for studying smaller areas. Their low air speed and high manoeuvrability enables them to sample the atmosphere with high spatial resolution. Equipped with an analyser for carbon dioxide UAS could also become a powerful validation tool for efforts to model dispersion of tracers on fine scales, e.g. inside street canyons.

Due to their unique capabilities and low cost, we foresee that the use of unmanned aircraft in the Earth sciences will significantly increase in the near future. We have shown how accurate measurements of the $CO_2$ dry air mole fraction can be taken on board small UAS and we anticipate these platforms to play an important role in closing gaps in the observation of the carbon cycle.

*Data availability.* The analyses presented here are based on many different experiments and use a combination of two or three different data sources in most cases. Compiling the data into a uniform, self-describing collection suitable for upload to a repository would be a great effort. Given the fact that our experiments were aimed at characterising COCAP, reuse of the data by other groups seems unlikely. Hence, we did not upload our measurement data to a repository. However, data from individual experiments is available from the corresponding author upon request.

Data from the ICOS station Lindenberg can be requested from ICOS-D (http://www.icos-infrastruktur.de/en/mitarbeiter/atmosphaerenprogramm/).

5  *Competing interests.*  Christine Hummelgård, Hans Martin and Henrik Rödjegård work for SenseAir AB, the manufacturer of the HPP sensor family. The other authors declare that they have no conflict of interest.

*Acknowledgements.*  We thank Maksym Bryzgalov (SenseAir AB, Sweden) for helping with the configuration and integration of the $CO_2$ sensor. Hök Instruments AB (Sweden) kindly provided software for COCAP's data logger. We thank Wieland Jeschag and Till Fastnacht for adapting this software to our needs. We acknowledge Jürgen Kaulfuß, who designed and built the field calibration device. Calibration
10  standards were prepared by the gas lab of the MPI for Biogeochemistry Jena and we are grateful for their support. We thank Frank Beyrich, Matthias Lindauer, Udo Rummel and Marcus Schumacher (Deutscher Wetterdienst, Germany) for access to the Lindenberg station, technical support and data sharing. We gratefully acknowledge the authors of various open-source software packages that were used in the project and for the preparation of the manuscript, in particular GNU Octave (including the *optim* package), KiCad, gnuplot, GIMP, Inkscape, LyX and LaTeX. We thank the Max Planck Society for generous financial support. Parts ofM.K. this work have been funded byks COST (European cooperation in science and technology) and the German academic exchange service DAAD for funding.